# ePC: Fast and Deep Predictive Coding in Digital Simulation

Cédric Goemaere [1]  Gaspard Oliviers [2 3]  Rafal Bogacz [2 3]  Thomas Demeester [1]

## Abstract

Predictive Coding (PC) offers a brain-inspired alternative to backpropagation for neural network training, described as a physical system minimizing its internal energy. While ideally suited for analog implementation, such hardware does not exist yet, and thus, in practice, PC is predominantly *digitally simulated*, requiring excessive amounts of compute while struggling to scale to deeper architectures. This paper reformulates PC to overcome this hardware-algorithm mismatch. First, we uncover how the canonical state-based formulation of PC (sPC) is, by design, deeply inefficient in digital simulation, inevitably resulting in exponential signal decay that stalls the entire numerical process. Then, to overcome this fundamental limitation, we introduce error-based PC (ePC), a novel reparameterization of PC which does not suffer from signal decay. Though no longer directly implementable in analog, ePC numerically computes *exact* PC weights gradients and runs orders of magnitude faster than sPC. Experiments across multiple architectures and datasets demonstrate that ePC matches backpropagation's performance even for deeper models where sPC struggles. Besides practical improvements, our work provides theoretical insight into PC dynamics and establishes a foundation for scaling PC-based learning to deeper architectures in digital simulation and beyond.

## 1. Introduction

Originally a neuroscience theory of cortical function (Friston & Kiebel, 2009), Predictive Coding (PC) has recently been reformulated as a general machine learning algorithm, offering a bio-inspired alternative to backpropagation with distinct learning dynamics (Bogacz, 2017; Whittington & Bogacz, 2017, 2019; Millidge et al., 2022a). Unlike backprop, PC produces weight gradients through a two-step process: first, infer the optimal state of neuron activations that should result from learning, and only then update the weights. This approach of "inferring activity before plasticity" (Song et al., 2024) has been found advantageous for learning: it improves the geometry of the loss landscape (Innocenti et al., 2024a) and reduces interference between competing training signals, leading to improved capabilities in online and continual learning settings (Song et al., 2024).

Our work focuses on PC's critical first step, inferring the optimal activations, specified as an energy minimization. Concretely, each layer tries to predict the state of the next layer, continually adjusting its own state to reduce the local prediction loss (known as 'energy'). Reminiscent of a physical process, PC would, in theory, be ideally suited for an analog implementation, promising fast and ultra-energy-efficient AI training. However, no such hardware exists yet, and instead, current PC research must rely on digital simulation with numerical solvers, requiring numerous iterations to reach state convergence. Due to this hardware-algorithm mismatch, PC incurs substantial overhead compared to backpropagation, which maps naturally to digital hardware.

Another scaling issue, observed by Pinchetti et al. (2025), is that, even in simple supervised settings, deeper PC-trained models often perform worse than shallower ones, in contrast to backprop. Recent efforts have explored this depth scaling failure from different angles. Several observed a highly uneven energy distribution across the network (Pinchetti et al., 2025; Qi et al., 2025; Ha et al., 2026), leading to weaker weight gradients for deeper layers; however, the underlying mechanism remained unclear. Proposed solutions either modify PC's weight gradient formulas (Qi et al., 2025), impose a fixed-prediction assumption (Ha et al., 2026) or stay limited to residual architectures (Innocenti et al., 2025). A general solution for standard feedforward models using exact PC remains an open problem.

In this paper, we address the fundamental limitations of PC's digital simulation—which currently represents nearly all practical PC research. Our work connects the seemingly disparate problems of depth scaling and computational efficiency in PC networks, uncovering a common underlying cause and providing a simple solution.

[1]IDLab, Ghent University – imec, Belgium [2]Brain Network Dynamics Unit, University of Oxford, UK [3]MRC CoRE in Restorative Neural Dynamics, UK. Correspondence to: Cédric Goemaere <cedric.goemaere@ugent.be>.

*Proceedings of the 43rd International Conference on Machine Learning*, Seoul, South Korea. PMLR 306, 2026. Copyright 2026 by the author(s).

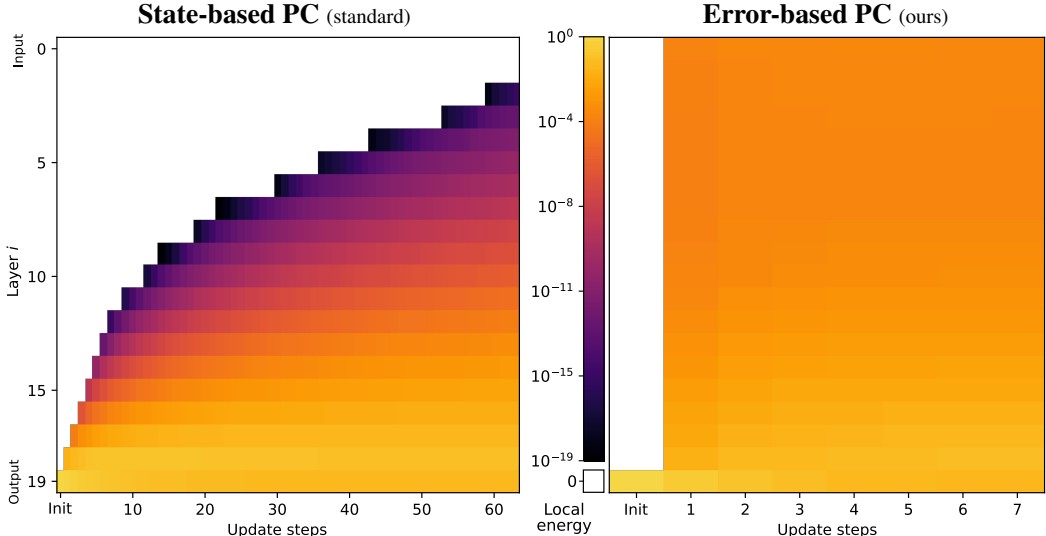

**Figure 1. Dynamics of layerwise energies in Predictive Coding.** Standard state-based PC struggles to propagate signals through the network (from output to input), with progressively longer delays at deeper layers. By contrast, error-based PC converges to equilibrium in just a few update steps, thanks to its global signal propagation. Results for an untrained 20-layer MLP on a random MNIST input.

**Our contributions**

- We identify a fundamental signal decay mechanism in traditional state-based PC (sPC), whereby signals attenuate exponentially as they propagate through the network (see Fig. 1), explaining both slow convergence and poor performance in deeper PC network simulations.

- We introduce error-based PC (ePC), a novel reparameterization of PC that eliminates signal decay by directly optimizing over prediction errors rather than faithfully simulating the physical process. Crucially, ePC provably computes the exact same state equilibrium as sPC.

- We empirically show that ePC converges up to three orders of magnitude faster than sPC on deep networks, resolving a major computational bottleneck in PC research.

- Through comprehensive experiments across architectures of varying depths, mirroring the setup of Pinchetti et al. (2025), we demonstrate that ePC consistently achieves performance comparable to backprop, providing an effective solution to PC's depth scaling issues in simulation.

**Scope and motivation**   While PC originated as a neuroscience theory, our work focuses solely on its application as a machine learning algorithm. Enforcing strict locality is essential for physical realizations (like the brain), but not for simulations, where it leads to slow convergence, as we will show. By relaxing this constraint, ePC is able to offer much faster PC simulations, finally enabling the iterative digital verification needed to commence analog chip development.

## 2. A Primer on Predictive Coding

As foundation for our contributions, we first present the canonical formulation of PC as an energy minimization over neural states. We will refer to this as **state-based PC (sPC)**. Note that sPC is just one possible instantiation of PC; while this distinction is often missing from literature, it is crucial to our premise.

In sPC (and PC in general), each layer attempts to predict the state of the next layer. The main goal is to minimize the energy function $E$, the sum of all prediction errors:

$$E(\boldsymbol{s}, \boldsymbol{\theta}) = \frac{1}{2} \sum_{i=0}^{L-1} \|\boldsymbol{s_i} - \hat{\boldsymbol{s}}_i\|^2 + \mathcal{L}(\hat{\boldsymbol{y}}, \boldsymbol{y}) \qquad (1)$$

$\boldsymbol{s_i}$ denotes the neural state at layer $i$ and $\hat{\boldsymbol{s}}_i := \boldsymbol{f}_{\boldsymbol{\theta}_i}(\boldsymbol{s_{i-1}})$ the parametrized prediction of $\boldsymbol{s_i}$ based on the preceding layer's state $\boldsymbol{s_{i-1}}$, as illustrated in Fig. 2a.

For ease of notation, we define $\boldsymbol{s_{-1}} := \boldsymbol{x}$ (the input data) and $\hat{\boldsymbol{y}} := \hat{\boldsymbol{s}}_L$ (output prediction of the target $\boldsymbol{y}$). The output loss $\mathcal{L}$ may be chosen freely (Pinchetti et al., 2022), with squared error being the common choice in PC literature.

As a learning algorithm, PC's primary purpose is to produce informative gradients for training the parameters $\boldsymbol{\theta}$. Contrary to backprop, sPC achieves this through purely local weight updates, relying on intermediaries (like the states) to spread the relevant learning signals across the network.

*(a)* State-based PC, the standard formulation of PC with a locally connected computational graph

*(b)* Error-based PC, our reparametrization of PC with a fully-connected (global) graph structure

*Figure 2.* Structural comparison of sPC (left) and ePC (right), highlighting functional equivalence

A single weight update step in sPC consists of a two-phased energy minimization of $E(s, \theta)$:

1. **State updates:** With the parameters $\theta$ kept fixed, the states $s$ evolve continuously to minimize $E$, until equilibrium is reached. The state dynamics for layer $i$ follow:

$$\frac{\partial s_i}{\partial t} = -\nabla_{s_i} E(s, \theta) = -\epsilon_i + \left(\frac{\partial f_{\theta_{i+1}}}{\partial s_i}\right)^T \epsilon_{i+1}, \quad (2)$$

where $\epsilon_i := s_i - \hat{s}_i$ represents the local prediction error.

2. **Weight update:** With $s$ kept fixed, the parameters $\theta$ are updated once, further minimizing $E$:

$$\Delta\theta_i \propto -\nabla_{\theta_i} E(s, \theta) = \left(\frac{\partial f_{\theta_i}}{\partial \theta_i}\right)^T \epsilon_i. \quad (3)$$

Full training involves repeating this process over numerous data batches, as in standard Deep Learning. The distinctive feature of sPC, however, is that both phases can be implemented efficiently in brain-like circuits with strictly local computation (Whittington & Bogacz, 2017, 2019), be it biological or electronic.

**Finding the state equilibrium** Notice how Eq. (3) requires only the final equilibrium states, discarding the trajectory taken to reach them. The specific method used to find these states is irrelevant to PC's weight updates, allowing researchers to freely choose their preferred approach.

For digital simulations, the most popular choice is to discretize Eq. (2) in time, reducing it to an SGD optimization

$$s_i \overset{\text{SGD}}{\Longleftarrow} s_i - \lambda\nabla_{s_i} E, \qquad \text{(State update step)}$$

with $\lambda$ the state learning rate (commonly, $\lambda \sim 0.01\text{-}0.1$). In practice, the number of update steps $T$ is often kept constant (a hyperparameter), and convergence is simply assumed.

The PC community has also experimented with more advanced simulation options. Some have looked into ODE

solvers (Innocenti et al., 2024b) or momentum-based optimizers (Pinchetti et al., 2025). Others have explored approximate one-step regimes (Salvatori et al., 2024), sequential update orders (Alonso et al., 2024), or auxiliary networks for direct equilibrium prediction (Tschantz et al., 2023). Yet, despite these advances, substantial gaps remain in computational efficiency and overall understanding of PC.

**Feedforward state initialization** A common practice is to set the initial states $s^{t=0}$ via a feedforward pass of the input $x$ through the network. At each layer, the prediction $\hat{s}_i$ is copied onto $s_i$, initializing the local energy $E_i$ to exactly zero. Despite lacking theoretical justification and physical implementability, this numerical technique is widely used due to its empirical success in accelerating the simulated state optimization process.

## 3. The Problem of Exponential Signal Decay in State-based Predictive Coding

In this section, we uncover a previously unidentified mechanism in state-based PC networks: the exponential decay of training signal during energy minimization. This discovery represents a fundamental limitation that affects the scalability of deep PC network simulations and helps explain the performance gap with backprop, which was observed to worsen for deeper models (Pinchetti et al., 2025).

### 3.1. Observation: Signals Do Not Propagate Smoothly

In sPC simulations, after feedforward state initialization, all energies are set to zero except for the output loss $\mathcal{L}$. Next, state updates should drive a backward signal from output to input, advancing one layer per update step.

The theory suggests a clear chain reaction: non-zero energy at any layer should induce changes in neighboring states, thereby continuously propagating the signal further down the network. However, our empirical observations do not reflect this expected behavior.

Fig. 1 illustrates how, in practice, the signal travels discontinuously through the network, halting at deeper layers with progressively longer delays.

Paradoxically, we observe that a non-zero energy at one layer fails to immediately propagate to adjacent layers, remaining dormant for multiple update steps before inducing detectable changes. This behavior seems to scale logarithmically with time, requiring exponentially many update steps for signals to reach the bottom layers—an impractical computational requirement.

### 3.2. Uncovering the Cause: a Problematic Mechanism of Exponential Signal Decay in sPC

To gain some insight into the cause of this suppressed signal propagation, we can track the state dynamics at the start of sPC. Below, in Fig. 3, we present a detailed, step-by-step description of the initial wavefront travelling backwards through the network.

Our analysis uncovers a signal decay mechanism: when an energy gradient propagates from one layer to the next, it is attenuated by the state learning rate $\lambda$ (necessarily $< 1$ for stability). With each subsequent layer traversal, this attenuation compounds multiplicatively, resulting in exponential decay with respect to network depth.

Specifically, for a network of depth $L$, the first non-zero signal to reach state $s_i$ can be modelled as

$$\Delta s_i \propto \lambda^{L-i} \nabla_{\hat{y}} \mathcal{L}, \tag{4}$$

with $\nabla_{\hat{y}} \mathcal{L}$ the initial loss gradient and the $\propto$-sign used to ignore layer effects, both pathological (vanishing/exploding) and corrective (skip connections/normalisation).

Crucially, this implies that the signal decay is unavoidable, even in carefully crafted, hyper-stable architectures like $\mu$PC (Innocenti et al. (2025); as demonstrated in Appendix D.1).

Moreover, for typical values of $\lambda$ (0.01-0.1), signals fall below numerical precision bounds within just 4 to 8 update steps.[1] This explains why theoretically continuous propagation manifests as discrete, delayed jumps in practice, with increasingly pronounced effects at greater network depths.

---

[1] In float32, addition only works up to 8 orders of magnitude (e.g., $1 + 10^{-8} = 1$), a concept known as "machine epsilon".

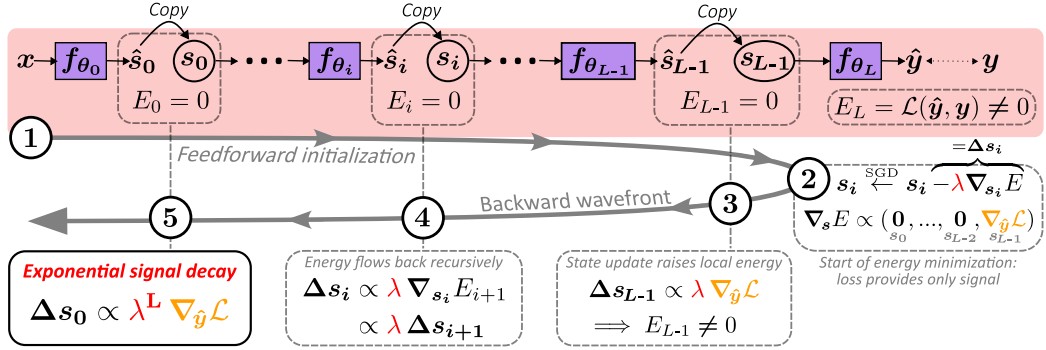

*To reduce clutter, layer effects are folded into the $\propto$-sign, which indicates matrix proportionality.*

**Step-by-step dynamics of state-based Predictive Coding for input $x$ and target $y$**

① **Feedforward initialization:** Due to state copying, all internal energies $E_0, \ldots, E_{L-1}$ start at zero, with in general a non-zero output energy $E_L$.

② **Start of energy minimization:** As gradient descent (with state learning rate $\lambda$) begins, the output loss introduces the only non-zero gradient $\nabla_{\hat{y}} \mathcal{L}$.

③ **Top-layer state update:** The top layer is updated first, raising its energy above zero.

④ **Recursive propagation:** A backward wavefront emerges: each non-zero energy induces a state change in the preceding layer, recursively propagating a diminishing signal.

⑤ **Exponential decay:** By the input layer, the signal has faded exponentially with depth $L$.

*Figure 3.* Step-by-step dynamics of sPC reveals an exponential signal decay in the backward path.

Persisting beyond just the initial wavefront, the signal decay mechanism plagues the entire energy minimization process. Appendix B provides an approximate global analysis, hinting at a hardware-algorithm mismatch where *digital simulations* of sPC are problematic, not sPC itself. While physical realizations (like the brain) would handle local interactions efficiently, enforcing these constraints digitally leads to signal decay and potentially misleading conclusions on (s)PC.

### 3.3. Implications for Training Deep sPC Networks

This exponential signal decay has profound implications for the digital simulation of sPC networks. Critically, **deeper layers may remain entirely untrained** if optimization is terminated before any signal has arrived, with these layers now effectively representing purely random input transformations. Such incomplete training would be hard to detect from state convergence metrics alone, which may incorrectly suggest equilibrium has been reached, when in reality, these layers have yet to begin meaningful optimization.

A more nuanced implication emerges when considering which signals do successfully penetrate the network. Only hard-to-classify or mislabeled inputs could produce output gradients $\nabla_{\hat{y}}\mathcal{L}$ that are large enough to overcome the exponential attenuation, potentially creating a systemic bias where different layers of the feedforward network train on different subsets of the data distribution.

Even when signals eventually reach deeper layers, the ensuing state modification will struggle to propagate back to upper layers. This creates a persistent misalignment where top layers, despite receiving strong output signals, cannot efficiently adapt to changes in deeper representations. While feedforward state initialization partially mitigates this issue, it cannot eliminate the intrinsic interdependencies that exist between states throughout optimization.

This signal propagation challenge represents a significant theoretical and practical limitation to scaling sPC networks in digital simulation. Common remedies, like increased learning rates (Pinchetti et al., 2025), higher numerical precision (Appendix B), careful weight initialization (Appendix D.1) or alternative optimizers (Pinchetti et al., 2025), all address symptoms without resolving the core issue, underscoring the need to identify a more suitable standard formulation of PC.

## 4. Shifting from States to Errors: Predictive Coding without Signal Decay

We introduce *error-based PC* (**ePC**), a novel reparameterization of PC that directly addresses the exponential signal decay problem identified in the previous section.

The key insight of ePC is to reformulate PC dynamics in terms of errors rather than states. By restructuring the computational graph from locally to globally connected, ePC enables signals to reach all layers simultaneously without attenuation.

While no longer directly implementable in a physical substrate (e.g., an analog circuit), ePC forms a helpful simulation tool that provably computes the same state equilibrium as sPC, resulting in exact-PC weight gradients. This means one may use ePC to signficantly speed up digital verification simulations of sPC circuits, paving the way towards practically viable analog hardware development for PC.

### 4.1. Mathematical Formulation of Error-Based PC

ePC reparameterizes PC by making the prediction errors $\epsilon$ the primary variables to optimize, rather than the states $s$.

The energy function remains the same, now formulated as:

$$E(\boldsymbol{\epsilon}, \boldsymbol{\theta}) = \frac{1}{2}\sum_{i=0}^{L-1}\|\boldsymbol{\epsilon}_i\|^2 + \mathcal{L}(\hat{\boldsymbol{y}}, \boldsymbol{y}) \text{ with } \hat{\boldsymbol{y}} = \boldsymbol{f_\theta}(\boldsymbol{x}, \boldsymbol{\epsilon}) \quad (5)$$

The core dynamics remain unchanged: during training, errors $\epsilon$ are iteratively updated to minimize $E$, followed by a gradient step to further minimize $E$ with respect to $\boldsymbol{\theta}$ (exactly Eq. (3) again). Crucially, ePC remains a valid PC algorithm (as technically verified in Appendix C.1).

When needed, states can be derived from errors through the recursive relationship $\boldsymbol{s}_i := \hat{\boldsymbol{s}}_i + \boldsymbol{\epsilon}_i$, where still $\hat{\boldsymbol{s}}_i := \boldsymbol{f}_{\boldsymbol{\theta}_i}(\boldsymbol{s}_{i-1})$. Conceptually, this amounts to a feedforward pass starting from the input $\boldsymbol{x}$ with perturbations $\boldsymbol{\epsilon}_i$ applied at each layer, as graphically shown in Fig. 2b.

Fig. 4 demonstrates the close algorithmic parallels between sPC and ePC, with a more extensive comparison given in Fig. A.1. Such strong similarities should not be surprising, as both methods are valid parametrizations of PC; in fact, they are equivalent (see proof in Appendix C.2).

### 4.2. How ePC Avoids Signal Decay

The key difference between sPC and ePC lies in the structure of their computational graph, as shown in Fig. 2. Striving for biological plausibility, sPC intentionally breaks the graph to enforce local update information, inadvertently resulting in exponential signal decay when simulated numerically, as explained in Section 3.

To avoid this issue, ePC reconnects the entire network graph, thereby creating a direct relationship between all input variables and the predicted output:

$$\textbf{(ePC)} \quad \hat{\boldsymbol{y}} = \text{func}(\boldsymbol{x}, \boldsymbol{\epsilon}_0, \boldsymbol{\epsilon}_1, \dots, \boldsymbol{\epsilon}_{L-1})$$

$$\text{vs. } \textbf{(sPC)} \quad \hat{\boldsymbol{y}} = \text{func}(\boldsymbol{s}_{L-1})$$

**Algorithm 1:** **State-based PC** *(standard)*

*State updates*

1: Initialize states $\{s_i\}$ $\leftarrow$ `ff_init`$(x)$
2: **for** $t = 1$ to $T$ **do**
3:    $s_{-1} \leftarrow x$
4:    **for** $i = 0$ to $L-1$ **do**        ▷ Parallel
5:       $\hat{s}_i \leftarrow f_{\theta_i}(s_{i-1})$
6:       $\epsilon_i \leftarrow s_i - \hat{s}_i$
7:    $\hat{y} \leftarrow f_{\theta_L}(s_{L-1})$
8:    $E \leftarrow \frac{1}{2}\sum_{i=0}^{L-1}\|s_i - \hat{s}_i\|^2 + \mathcal{L}(\hat{y}, y)$
9:    $\nabla_{s_j}E \leftarrow \epsilon_j - \frac{\partial \hat{s}_{j+1}}{\partial s_j}^T \epsilon_{j+1}$    ▷ Local
10:    $s_j \leftarrow s_j - \lambda \nabla_{s_j}E$ for all $j$

*Weight update*

11: $\nabla_{\theta_j}E \leftarrow -\frac{\partial \hat{s}_j}{\partial \theta_j}^T \epsilon_j$    ▷ Local
12: $\theta_j \leftarrow \theta_j - \eta \nabla_{\theta_j}E$ for all $j$

**Algorithm 2:** **Error-based PC** *(ours)*

*Error updates*

1: Initialize errors $\{\epsilon_i\}$ $\leftarrow$ `zero_init`
2: **for** $t = 1$ to $T$ **do**
3:    $s_{-1} \leftarrow x$
4:    **for** $i = 0$ to $L-1$ **do**       ▷ Sequential
5:       $\hat{s}_i \leftarrow f_{\theta_i}(s_{i-1})$
6:       $s_i \leftarrow \hat{s}_i + \epsilon_i$
7:    $\hat{y} \leftarrow f_{\theta_L}(s_{L-1})$
8:    $E \leftarrow \frac{1}{2}\sum_{i=0}^{L-1}\|\epsilon_i\|^2 + \mathcal{L}(\hat{y}, y)$
9:    $\nabla_{\epsilon_j}E \leftarrow \epsilon_j + \frac{\partial \hat{y}}{\partial \epsilon_j}^T \nabla_{\hat{y}}\mathcal{L}$   ▷ Reverse-mode AD
10:    $\epsilon_j \leftarrow \epsilon_j - \lambda \nabla_{\epsilon_j}E$ for all $j$

*Weight update*

11: $\nabla_{\theta_j}E \leftarrow -\frac{\partial \hat{s}_j}{\partial \theta_j}^T \epsilon_j$    ▷ Local
12: $\theta_j \leftarrow \theta_j - \eta \nabla_{\theta_j}E$ for all $j$

*Figure 4.* Algorithmic comparison of sPC vs. ePC, with structural differences highlighted in color. Loops over $j$ are omitted for brevity. An extended version is provided in Appendix A.

This restructuring enables the main advantage of ePC: the use of *reverse-mode automatic differentiation (AD)*[2] to transmit signals from the output loss $\mathcal{L}(\hat{y}, y)$ directly to all errors $\epsilon_i$ via $\hat{y}$, without intermediate attenuation.

A brief step-by-step analysis reveals how ePC successfully decouples stability from propagation speed, which were problematically intertwined in sPC. First, reverse-mode AD computes gradients throughout the entire network, ensuring signals reach all layers unattenuated. Only thereafter, during the actual error update step, is the learning rate applied, affecting stability but not propagation reach. This separation allows signals to influence all network layers simultaneously, regardless of depth, thereby eliminating the exponential decay problem seen in sPC.

While ePC might appear to be a hybrid of PC and backprop, this characterization is misleading: ePC remains fundamentally a PC algorithm. Reverse-mode AD serves only as a computational backbone to efficiently reach state equilibrium in digital simulation, without influencing the weight updates, which stay temporally local following PC principles. Appendices C.3 and C.4 explore the nuanced relationship between ePC and backpropagation in greater detail.

---

[2]We use "reverse-mode AD" to refer to the efficient numerical method to compute gradients of a parametrized function. Though often used interchangeably, we reserve "backprop(agation)" for the learning algorithm that *uses* these gradients for parameter updates.

### 4.3. Proof-of-Concept on MNIST

To evaluate the practical advantages of ePC over sPC, we compared the two methods for a 20-layer linear network trained on MNIST.

This architecture forms an ideal testbed as it is reasonably deep and offers a unique, analytically tractable equilibrium solution that can serve as ground truth. We picked a simple task to match the network's limited representational capacity. For an unbiased comparison, we used identical network weights for both methods (obtained through backpropagation as neutral learning method), with hyperparameters optimized for convergence speed. Complete details are provided in Appendix E.1.

Fig. 5 illustrates how both sPC and ePC converge to the analytical optimum, reconfirming their theoretical equivalence. However, for the exact same model, ePC converges over $100\times$ faster than sPC, a huge difference in speed that highlights ePC's practical advantage for training deep PC networks on digital hardware.

The figure also provides further evidence for the discontinuous signal propagation issue identified in Section 3. In sPC, the signal takes roughly 30 steps to advance 9 layers and reach $s_9$, and nearly 100 steps to traverse the 20-layer network and reach $s_0$. In contrast, ePC has long converged by then, with its global connectivity enables all layers to optimize immediately and simultaneously.

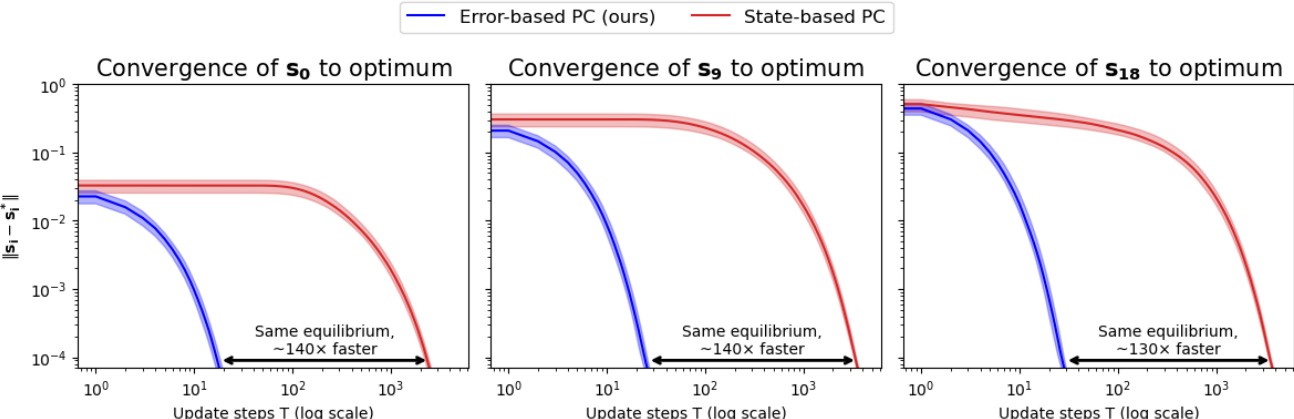

*Figure 5.* State convergence dynamics for the bottom, middle, and top hidden layers in a 20-layer linear PC network trained on MNIST. Curves show batch medians (n=64) of L2 distance to the analytical optimum, with interquartile shading. Thanks to its superior signal propagation, ePC converges over $100\times$ faster than sPC, while reaching the exact same equilibrium.

Additional experiments with deep non-linear MLPs (see Appendix D.2) yielded similar results, with ePC consistently outperforming sPC in terms of convergence speed. Notably, sPC required an impractical number of update steps (>100,000) to actually reach proper convergence, reinforcing the necessity of our ePC reformulation for scaling PC to deeper networks.

### 4.4. Implications for Deep PC Networks

The benefits of ePC's improved signal propagation extend beyond faster convergence, addressing several fundamental limitations of sPC. Most importantly, by providing signal to all layers from the first step, ePC completely resolves the issue of untrained deep layers. It enables all layers to begin optimization simultaneously, regardless of network depth.

Furthermore, ePC eliminates the potential systemic bias in sPC, where only inputs generating large output gradients could successfully influence deeper layers. With ePC, all inputs contribute equally to training at all depths, promoting uniform learning across the network.

Moreover, any change in deeper layers is efficiently communicated to the upper layers through the feedforward pass required for $\hat{y}$. This bidirectional efficiency explains why the widely-used feedforward state initialization heuristic in PC works so well: it essentially implements the first step of ePC. Our formulation thus provides theoretical backing for this empirical practice while extending its benefits throughout the optimization process.

By resolving these fundamental limitations, ePC establishes a solid foundation for scaling PC to deeper architectures. Our MNIST proof-of-concept demonstrates significant con-

vergence improvements, validating this theoretical advancement and motivating large-scale empirical evaluation.

## 5. Experiments

To evaluate ePC's effectiveness in training deep networks and compare it against sPC, we conducted extensive experiments using backprop as gold standard. Our experimental design follows the community's established benchmark from Pinchetti et al. (2025), allowing direct comparison with their findings.

### 5.1. Experimental Setup

We evaluated performance across four standard computer vision datasets: MNIST (LeCun, 1998; Cohen et al., 2017), FashionMNIST (Xiao et al., 2017), and CIFAR-10/100 (Krizhevsky, 2009). The architecture selection spanned an MLP, VGG-style convolutional networks of various depths (Simonyan & Zisserman, 2014), and a deep residual network (He et al., 2016). The output loss $\mathcal{L}$ is either Mean Squared Error (MSE) or Cross-Entropy (CE), again mirroring Pinchetti et al. (2025).

Complete implementation details, including hyperparameter settings, are provided in Appendix E. All code is available at https://github.com/cgoemaere/error_based_PC.

### 5.2. Results and Analysis

Our results in Table 1 confirm the significant performance gap between sPC and backpropagation previously reported by Pinchetti et al. (2025), while demonstrating that ePC substantially narrows this gap.

*Table 1.* Test accuracies (in %) of ePC, sPC and backprop for various models, losses, and datasets. Bold indicates best results within confidence intervals (mean ± 1 std. dev.; taken over 5 seeds).

| Loss $\mathcal{L}$ | Mean Squared Error | | | Cross-Entropy | | |
|---|---|---|---|---|---|---|
| Training algorithm | ePC | sPC | Backprop | ePC | sPC | Backprop |
| **MLP (4 layers)** | | | | | | |
| MNIST | $\mathbf{98.28^{\pm 0.09}}$ | $\mathbf{98.42^{\pm 0.08}}$ | $\mathbf{98.30^{\pm 0.15}}$ | $\mathbf{98.11^{\pm 0.08}}$ | $98.01^{\pm 0.15}$ | $\mathbf{98.13^{\pm 0.08}}$ |
| FashionMNIST | $87.02^{\pm 0.24}$ | $88.01^{\pm 0.09}$ | $\mathbf{88.79^{\pm 0.21}}$ | $87.58^{\pm 0.13}$ | $88.00^{\pm 0.24}$ | $\mathbf{88.87^{\pm 0.27}}$ |
| **VGG-5** | | | | | | |
| CIFAR-10 | $\mathbf{88.70^{\pm 0.12}}$ | $86.67^{\pm 0.20}$ | $\mathbf{88.58^{\pm 0.12}}$ | $\mathbf{88.27^{\pm 0.18}}$ | $84.66^{\pm 0.33}$ | $\mathbf{87.95^{\pm 0.29}}$ |
| CIFAR-100 (Top-1) | $64.37^{\pm 0.17}$ | $50.41^{\pm 1.45}$ | $\mathbf{64.80^{\pm 0.24}}$ | $63.39^{\pm 0.25}$ | $56.85^{\pm 0.69}$ | $\mathbf{63.83^{\pm 0.15}}$ |
| CIFAR-100 (Top-5) | $85.28^{\pm 0.38}$ | $77.41^{\pm 1.21}$ | $\mathbf{85.80^{\pm 0.13}}$ | $\mathbf{87.34^{\pm 0.14}}$ | $83.11^{\pm 0.19}$ | $\mathbf{87.43^{\pm 0.06}}$ |
| **VGG-7** | | | | | | |
| CIFAR-10 | $\mathbf{88.98^{\pm 0.19}}$ | $77.79^{\pm 0.34}$ | $\mathbf{88.94^{\pm 0.32}}$ | $88.84^{\pm 0.31}$ | $77.98^{\pm 0.40}$ | $\mathbf{89.60^{\pm 0.16}}$ |
| CIFAR-100 (Top-1) | $\mathbf{66.55^{\pm 0.45}}$ | $42.90^{\pm 0.43}$ | $\mathbf{66.23^{\pm 0.42}}$ | $58.62^{\pm 0.20}$ | $53.45^{\pm 0.38}$ | $\mathbf{65.14^{\pm 0.29}}$ |
| CIFAR-100 (Top-5) | $\mathbf{85.65^{\pm 0.12}}$ | $70.01^{\pm 0.52}$ | $84.10^{\pm 0.39}$ | $85.09^{\pm 0.14}$ | $80.48^{\pm 0.38}$ | $\mathbf{88.60^{\pm 0.24}}$ |
| **VGG-9** | | | | | | |
| CIFAR-10 | $88.80^{\pm 0.71}$ | $76.40^{\pm 0.20}$ | $\mathbf{90.04^{\pm 0.50}}$ | $86.81^{\pm 0.09}$ | $78.60^{\pm 0.30}$ | $\mathbf{89.76^{\pm 0.20}}$ |
| CIFAR-100 (Top-1) | $61.35^{\pm 0.76}$ | $45.70^{\pm 0.14}$ | $\mathbf{66.28^{\pm 0.29}}$ | $\mathbf{60.65^{\pm 0.25}}$ | $54.19^{\pm 0.41}$ | $\mathbf{61.11^{\pm 0.45}}$ |
| CIFAR-100 (Top-5) | $\mathbf{84.74^{\pm 0.40}}$ | $73.04^{\pm 0.46}$ | $\mathbf{84.96^{\pm 0.29}}$ | $\mathbf{85.84^{\pm 0.15}}$ | $80.65^{\pm 0.41}$ | $85.14^{\pm 0.32}$ |
| **ResNet-18** | | | | | | |
| CIFAR-10 | $\mathbf{92.17^{\pm 0.26}}$ | "$53.74^{\pm 0.43}$" | $\mathbf{92.36^{\pm 0.12}}$ | $\mathbf{91.73^{\pm 0.21}}$ | "$43.19^{\pm 0.61}$" | $\mathbf{91.85^{\pm 0.24}}$ |
| CIFAR-100 (Top-1) | $68.52^{\pm 0.34}$ | "$22.83^{\pm 0.38}$" | $\mathbf{69.94^{\pm 0.54}}$ | $69.47^{\pm 0.32}$ | "$16.01^{\pm 0.42}$" | $\mathbf{71.46^{\pm 0.32}}$ |
| CIFAR-100 (Top-5) | $86.86^{\pm 0.44}$ | "$50.18^{\pm 0.52}$" | $\mathbf{87.76^{\pm 0.41}}$ | $90.47^{\pm 0.12}$ | "$40.67^{\pm 0.70}$" | $\mathbf{91.91^{\pm 0.23}}$ |

"...": *ResNet-18 was unstable in our sPC experiments, so we copied the results from Pinchetti et al. (2025)*

Several key findings emerge from our experiments:

- **ePC scales well to deeper networks**, attaining ever improved accuracies similar to backpropagation and unlike sPC, which degraded with network depth. This is most noticeable for ResNet-18, where ePC achieved competitive performance while sPC suffered from instability issues in our implementation.

- **ePC matches backpropagation's results** across most datasets and architectures, with results often falling within statistical confidence bounds.

- **These trends hold across both loss functions** in our setup, despite CE's typically superior gradient properties. However, we did observe greater sensitivity to hyperparameter selection with CE loss in both ePC and sPC.

Overall, our experimental results validate ePC's theoretical advantages. By resolving sPC's signal decay problem, ePC successfully scales PC to deeper architectures, unlocking its ability to handle substantially more complex machine learning challenges than previously possible.

# 6. Conclusion and Future Directions

This paper identifies and addresses a fundamental limitation in PC simulations: exponential signal decay during state-based energy minimization. Our proposed error-based formulation overcomes this limitation by restructuring PC's computational graph while preserving theoretical equivalence, achieving dramatic performance improvements that finally establish PC as a competitive alternative to backprop for training deep neural networks.

## 6.1. Reinterpreting PC as Minimal-Norm Perturbations

ePC provides a fresh perspective on PC's energy minimization process. Essentially, (e)PC searches for minimal-norm layerwise perturbations that collectively produce optimal outputs. At each layer, these corrections are added to the feedforward pass, incrementally refining the final output prediction. From these targeted state modifications, local weight learning rules can then be derived.

This reframing connects naturally with the Least-Control Principle (Meulemans et al., 2022), in which an external controller tries to minimally steer network activities to pro-

duce the target output. In their Appendix S4, they briefly explore PC through the lens of control theory, identifying the errors as an optimal control. With their framework allowing arbitrary controller circuits, it may be possible to find a biologically plausible implementation of ePC that does not explicitly require reverse-mode AD, thereby addressing what some may consider essential for a PC algorithm.

## 6.2. Predictive Coding Beyond The Hardware Lottery

Algorithmic success is often dictated not by theoretical merit but by compatibility with prevailing hardware (Hooker, 2021). Serving as a prime example, PC has struggled to prove its worth despite theoretical soundness. To unlock its full potential, ePC reformulates PC in a way that aligns naturally with digital processors, relying on reverse-mode AD to efficiently spread signals across deep networks. Meanwhile, sPC remains highly relevant for neuromorphic implementations, where physical energy minimization occurs naturally and near-instantaneously, regardless of network depth.

Despite their structural differences, both approaches still minimize the same energy function to reach identical equilibria. This functional equivalence creates a pragmatic research methodology: rather than being limited by sPC's digital inefficiency, researchers can turn to ePC for rapid prototyping, generating insights that remain valid for understanding bio-plausible PC learning mechanisms.

## 6.3. The Road Ahead for Predictive Coding

With PC's viability as a learning algorithm now firmly established, research must shift from proof-of-concept to impact. We highlight two research directions with great potential:

1. **Unblocking neuromorphic hardware development:** Despite its theoretical suitability for ultra-energy-efficient neuromorphic implementation, hardware development for PC has been scarce. A key obstacle is our limited understanding of PC's behavior at exact equilibrium—the regime to which any physical implementation would naturally settle. While a recent analysis of this setting identified improved learning capabilities (Innocenti et al., 2024a), our experiments consistently preferred hyperparameter configurations of approximate backpropagation, leaving little appeal to hardware developers. With ePC as an efficient tool to further study equilibrium dynamics, research can finally begin to address this critical barrier to neuromorphic advancement.

2. **Identifying PC's distinctive advantages:** Rather than competing with backprop in its domains of strength, research should focus on areas where PC uniquely excels. As Song et al. (2024) demonstrated with online and continual learning, such domains exist but remain underexplored. Although ePC's reliance on reverse-mode AD

puts an upper limit on PC's computational efficiency (as noted before in Zahid et al., 2023), few-step ePC could offer a compromise that maintains PC's unique properties while keeping training times practical.

With ePC effectively addressing PC's computational limitations in digital simulation, the field must now face its true test: demonstrating that Predictive Coding offers substantive advantages in specific domains, sufficient to justify its adoption over established approaches.

*A limitations section is provided at the start of the appendix.*

## Reproducibility Statement

We took great care to ensure reproducibility, listing architectural details, hyperparameter sweep intervals, final values and even pseudorandom seeds, which can all be found in Appendix E. On the algorithmic level, Appendix A provides an extensive description of both sPC and ePC. Finally, our code is available at https://github.com/cgoemaere/error_based_PC.

## Acknowledgments and Disclosure of Funding

We are grateful to the anonymous reviewers and the T2K team at Ghent University for their valuable feedback on the manuscript, which helped improve the presentation, clarity, and accessibility of this paper. We also thank Francesco Innocenti for helpful discussions on signal decay in continuous-time systems that improved the correctness of Appendix B.

Final revisions were completed during CG's research visit to Prof. Bogacz's group in Oxford.

This research was partly funded by the Research Foundation – Flanders (FWO-Vlaanderen) through CG's PhD Fellowship (11PR824N) and grants G0C2723N and V439825N (the latter supporting the Oxford visit), as well as by the Flemish Government's AI Research Program. RB and GO were supported by the Medical Research Council (grants MC_UU_00003/1 and UKRI/MR/B000936/1) and Wellcome Trust (grant 313955/Z/24/Z).

## Impact Statement

This paper presents work whose goal is to advance the field of Machine Learning. There are many potential societal consequences of our work, none which we feel must be specifically highlighted here.

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

# Appendix

## Limitations

Our work demonstrates significant improvements in Predictive Coding efficiency and scalability. Nonetheless, certain limitations remain, which we discuss below.

**Different optimization trajectories**: While mathematically equivalent at equilibrium, sPC and ePC follow distinct optimization trajectories. Therefore, ePC cannot be used for research on the intermediate state dynamics of sPC (e.g., Millidge et al., 2024; Lee et al., 2025). Furthermore, considering the abundance of local minima present in deep neural networks, it is, in theory, possible that ePC and sPC may converge to different equilibria, though we did not observe any evidence of this during our experiments, not even for very deep MLPs (see Appendix D.2).

**Experimental scope**: Following the established PC benchmark by Pinchetti et al. (2025), we tested exclusively on standard supervised learning tasks (MNIST, FashionMNIST, CIFAR) where backprop is known to perform exceptionally well. The goal of our experiments was solely to demonstrate ePC's superiority over sPC, not to prove PC superiority over backpropagation. It would be valuable to explore ePC in domains where PC might have advantages, such as online and continual learning (Song et al., 2024), to determine whether these benefits extend to deeper architectures (now possible with ePC) or were simply artifacts of sPC's poor signal propagation.

## Clarifying Remarks

Certain aspects of our methodology may be misunderstood as limitations, but they instead reflect deliberate design choices and intrinsic advantages. For clarity, we highlight them here.

**Choice of baseline methods**: We compare ePC only against sPC and backprop (as neutral gold standard), rather than including a broad range of PC algorithms. This focused scope is intentional: Pinchetti et al. (2025) already extensively tested numerous PC variants and found that none successfully scaled to deep networks (see their Table 1). This is not surprising: these methods all build upon sPC, inheriting its signal decay problem. Thus, comparing against vanilla sPC (the most established and widely used PC algorithm) was the most appropriate choice.

**Biological plausibility**: Although ePC preserves PC's core theoretical foundations, its use of reverse-mode AD makes it unsuitable for direct biological implementation. However, ePC was never intended as a bio-plausible algorithm; instead, it serves as a computationally efficient tool for studying PC dynamics in digital simulation. In obeying physical constraints, sPC becomes impractical in simulation, as biological and digital systems operate under fundamentally different mechanisms. By turning PC into a practical numerical algorithm, ePC enables more extensive research into PC dynamics, potentially offering greater value to computational neuroscience than the biologically constrained but computationally intractable sPC.

**Computational efficiency**: In a persistent misconception, sPC is often touted for its parallelization abilities (Millidge et al., 2022a; Pinchetti et al., 2022; Salvatori et al., 2024; Pinchetti et al., 2025). However, this alleged advantage is fundamentally flawed. Even with perfect parallelization, PC networks with depth $L$ require at least $L$ sequential update steps because signals can only advance one layer per step due to local-only interactions (Zahid et al., 2023). In fact, our experiments demonstrate this to be a very loose lower bound: sPC requires *exponentially* many update steps to reach equilibrium, fully undoing any potential (linear) speed-up from parallelization. Moreover, it may prove difficult to actually parallelize layers with different dimensions, forcing sequential processing in practice (Pinchetti et al., 2025, Section 6.1). As a result, our PyTorch implementation of ePC takes only 5-20% longer per step compared to sPC, despite its strictly sequential nature. This minor cost is easily offset by ePC's exponential reduction in required steps, representing a massive net gain in computational efficiency. Of course, these comparisons of software implementations may ultimately be less relevant, as PC's true advantage lies in its suitability for ultra-fast neuromorphic hardware.

## A. Comparison of State-based vs. Error-based Predictive Coding

State-based PC (standard)

$x \rightarrow f_{\theta_0} \rightarrow \hat{s}_0 \; \fbox{$s_0$} \rightarrow f_{\theta_1} \rightarrow \hat{s}_1 \; \fbox{$s_1$} \rightarrow f_{\theta_2} \rightarrow \hat{y}$

$\epsilon_0 \qquad \epsilon_1$

$E = \frac{1}{2} \sum_i ||s_i - \hat{s}_i||^2 + \mathcal{L}(\hat{y}, y)$

$\dot{s}_i = -\nabla_{s_i} E \qquad \epsilon_i := s_i - \hat{s}_i$

Error-based PC (ours)

$x \rightarrow f_{\theta_0} \xrightarrow{\hat{s}_0} \oplus \xrightarrow{s_0} f_{\theta_1} \xrightarrow{\hat{s}_1} \oplus \xrightarrow{s_1} f_{\theta_2} \rightarrow \hat{y}$

$\epsilon_0 \qquad \epsilon_1$

$E = \frac{1}{2} \sum_i ||\epsilon_i||^2 + \mathcal{L}(\hat{y}, y)$

$\dot{\epsilon}_i = -\nabla_{\epsilon_i} E \qquad s_i := \hat{s}_i + \epsilon_i$

---

**Algorithm 3:** **State-based PC** *(standard)*

**Require:** Input $x$, target $y$, layers $\{f_{\theta_i}\}_{i=0}^{L}$, optimization steps $T$, state learning rate $\lambda$, weight learning rate $\eta$, output loss $\mathcal{L}$

*Feedforward state initialization* (ff_init)
1: $s_{-1} \leftarrow x$
2: **for** $i = 0$ to $L - 1$ **do**          ▷ Sequential
3: $\quad \hat{s}_i \leftarrow f_{\theta_i}(s_{i-1})$
4: $\quad s_i \leftarrow \hat{s}_i$

*State updates*
5: **for** $t = 1$ to $T$ **do**

6: $\quad$ **for** $i = 0$ to $L - 1$ **do**          ▷ Parallel
7: $\quad\quad \hat{s}_i \leftarrow f_{\theta_i}(s_{i-1})$
8: $\quad\quad \epsilon_i \leftarrow s_i - \hat{s}_i$
9: $\quad \hat{y} \leftarrow f_{\theta_L}(s_{L-1})$
10: $\quad E \leftarrow \frac{1}{2} \sum_{i=0}^{L-1} ||s_i - \hat{s}_i||^2 + \mathcal{L}(\hat{y}, y)$

*Local energy gradients w.r.t. states*
11: $\quad \epsilon_L \leftarrow \nabla_{\hat{y}} \mathcal{L}$
12: $\quad$ **for** $j = 0$ to $L - 1$ **do**          ▷ Parallel
13: $\quad\quad \nabla_{s_j} E \leftarrow \epsilon_j - \frac{\partial \hat{s}_{j+1}}{\partial s_j}^T \epsilon_{j+1}$
14: $\quad\quad s_j \leftarrow s_j - \lambda \nabla_{s_j} E$

*Local weight update*
15: **for** $j = 0$ to $L - 1$ **do**          ▷ Parallel
16: $\quad \nabla_{\theta_j} E \leftarrow -\frac{\partial \hat{s}_j}{\partial \theta_j}^T \epsilon_j$
17: $\quad \theta_j \leftarrow \theta_j - \eta \nabla_{\theta_j} E$

---

**Algorithm 4:** **Error-based PC** *(ours)*

**Require:** Input $x$, target $y$, layers $\{f_{\theta_i}\}_{i=0}^{L}$, optimization steps $T$, error learning rate $\lambda$, weight learning rate $\eta$, output loss $\mathcal{L}$

*Zero error initialization* (zero_init)
1: **for** $i = 0$ to $L - 1$ **do**          ▷ Parallel
2: $\quad \epsilon_i \leftarrow 0$

*Error updates*
3: **for** $t = 1$ to $T$ **do**
4: $\quad s_{-1} \leftarrow x$
5: $\quad$ **for** $i = 0$ to $L - 1$ **do**          ▷ Sequential
6: $\quad\quad \hat{s}_i \leftarrow f_{\theta_i}(s_{i-1})$
7: $\quad\quad s_i \leftarrow \hat{s}_i + \epsilon_i$
8: $\quad \hat{y} \leftarrow f_{\theta_L}(s_{L-1})$
9: $\quad E \leftarrow \frac{1}{2} \sum_{i=0}^{L-1} ||\epsilon_i||^2 + \mathcal{L}(\hat{y}, y)$

*Global reverse-mode AD w.r.t. errors*
10: $\quad \hat{s}_L \leftarrow \hat{y}$
11: $\quad$ **for** $j = L - 1$ to $0$ **do**          ▷ Sequential
12: $\quad\quad \nabla_{\epsilon_j} \mathcal{L} \equiv \nabla_{\hat{s}_j} \mathcal{L} \leftarrow \frac{\partial \hat{s}_{j+1}}{\partial \hat{s}_j}^T \nabla_{\hat{s}_{j+1}} \mathcal{L}$
13: $\quad$ **for** $j = 0$ to $L - 1$ **do**          ▷ Parallel
14: $\quad\quad \nabla_{\epsilon_j} E \leftarrow \epsilon_j + \nabla_{\epsilon_j} \mathcal{L}$
15: $\quad\quad \epsilon_j \leftarrow \epsilon_j - \lambda \nabla_{\epsilon_j} E$

*Local weight update*
16: **for** $j = 0$ to $L - 1$ **do**          ▷ Parallel
17: $\quad \nabla_{\theta_j} E \leftarrow -\frac{\partial \hat{s}_j}{\partial \theta_j}^T \epsilon_j$
18: $\quad \theta_j \leftarrow \theta_j - \eta \nabla_{\theta_j} E$

---

*Figure A.1.* Full side-by-side comparison of state-based PC (left) and error-based PC (right)

```
def ff_init(x):
  return [(x := f(x)) for f in layers[:-1]]

def get_E(s)
  s_pred = [f(s) for f,s in zip(layers, [x]+s)]
  s_pred, y_pred = s_pred[:-1], s_pred[-1]

  E = 0.5 * sum(
    L2norm(s_i-s_i_pred)**2
    for s_i, s_i_pred in zip(s, s_pred)
  )
  E += loss(y_pred, y)

def get_final_state():
  s = ff_init(x)
  s_optim = SGD(s, lr=lambda)
  for _ in range(T):
    s_optim.zero_grad()
    E = get_E(s)
    E.backward()
    s_optim.step()

def sPC_weight_update(w_optim):
  s = get_final_state()
  w_optim.zero_grad()
  E = get_E(s)
  E.backward()
  w_optim.step()
```

*(a)* State-based Predictive Coding

```
def states_from_errors(x, e):
  return [
    (x := f(x) + e_i).detach() # no backprop
    for f, e_i in zip(layers[:-1], e)
  ]

def zero_init():
  return [zeros(shape) for shape in shapes]

def y_pred(x):
  s_i = x
  for f, e_i in zip(layers, e + [0.0]):
    s_i = f(s_i) + e_i
  return s_i

def get_E_errors(e)
  E = 0.5 * sum(L2norm(e_i)**2 for e_i in e)
  E += loss(y_pred(x), y)

def get_final_errors():
  e = zero_init()
  e_optim = SGD(e, lr=lambda)
  for _ in range(T):
    e_optim.zero_grad()
    E = get_E_errors(e)
    E.backward()
    e_optim.step()

def ePC_weight_update(w_optim):
  e = get_final_errors()
  s = states_from_errors(x, e)
  w_optim.zero_grad()
  E = get_E(s)
  E.backward()
  w_optim.step()
```

*(b)* Error-based Predictive Coding

*Figure A.2.* PyTorch-style pseudocode comparison of sPC vs. ePC

## B. Signal Decay in State-based PC, Across Time and Hardware Constraints

This appendix extends our analysis of the exponential signal decay phenomenon identified in Section 3. While the main paper demonstrated how signals attenuate during the initial backward wavefront, we here derive a complete characterization of network dynamics that reveals the underlying mathematical structure governing signal propagation throughout energy minimization.

Our analysis uncovers a striking similarity to a simple binomial model, providing both theoretical insights into the discrete-time nature of the problem and practical understanding of why physical implementations of sPC (like the brain) would not suffer from the same limitations.

### B.1. Introducing a Simplified sPC Model for Analytical Tractability

To enable rigorous mathematical analysis, we introduce a simplified model that captures the essential dynamics while remaining analytically tractable. Note that this setting provides only a coarse approximation to the true state dynamics of sPC, in contrast to our exact analysis of the initial backward wavefront in Section 3.

**Key Assumption for Appendix B** *After feedforward state initialization, all state predictions $\hat{s}_i$ remain constant throughout energy minimization. This assumption implies that signal propagation occurs exclusively in the top-down direction, from output toward input layers.*

This simplification provides a reasonable approximation during early-stage optimization, where state dynamics are primarily driven by the output loss $\mathcal{L}$ before significant bottom-up signals emerge. However, it breaks down as the system evolves and predictions begin to change.

**Simplified Backward Dynamics** As described in Section 2, the temporal dynamics of states in sPC follow gradient descent on the energy function $E$ with state learning rate $\lambda$:

$$s_i^{t+1} = s_i^t - \lambda \boldsymbol{\nabla}_{s_i} E^t$$
$$= s_i^t - \lambda \epsilon_i^t + \lambda \epsilon_{i+1}^t \frac{\partial f_{\theta_{i+1}}}{\partial s_i}(s_i^t),$$

where $\epsilon_i := s_i - \hat{s}_i$ represents the layerwise prediction error. Given our key assumption above, this is equivalent to the deviation of each layer's state from its fixed prediction.

To further simplify our analysis, we set $\frac{\partial f_{\theta_{i+1}}}{\partial s_i}(s_i^t) \equiv I$, reducing the dynamics to:

$$s_i^{t+1} = s_i^t - \lambda \epsilon_i^t + \lambda \epsilon_{i+1}^t$$

## B.2. The Emergence of Recursive Error Dynamics Beyond the Wavefront

Since state predictions $\hat{s}_i$ remain fixed by assumption, the prediction errors $\epsilon_i$ follow the same temporal dynamics as the states themselves:

$$s_i^{t+1} = s_i^t - \lambda \epsilon_i^t + \lambda \epsilon_{i+1}^t$$
$$\Longleftrightarrow (s_i^{t+1} - \hat{s}_i) = (s_i^t - \hat{s}_i) - \lambda \epsilon_i^t + \lambda \epsilon_{i+1}^t$$
$$\Longleftrightarrow \epsilon_i^{t+1} = \epsilon_i^t - \lambda \epsilon_i^t + \lambda \epsilon_{i+1}^t$$
$$\Longleftrightarrow \epsilon_i^{t+1} = (1-\lambda)\epsilon_i^t + \lambda \epsilon_{i+1}^t$$

For small errors and/or learning rates, we can approximate the magnitude of the right-hand side as the sum of magnitudes, giving rise to recursive dynamics:

$$||\epsilon_i^{t+1}|| \approx (1-\lambda)||\epsilon_i^t|| + \lambda||\epsilon_{i+1}^t||$$

This recursive formula, when traced through the first few time steps, generates a striking pattern. Writing the magnitudes relative to the driving signal (i.e., the output gradient $\boldsymbol{\nabla}_{\hat{y}}\mathcal{L}$) gives us:

| Time | $t=0$ | $t=1$ | $t=2$ | $t=3$ | $t=0$ | $t=1$ | $t=2$ | $t=3$ |
|---|---|---|---|---|---|---|---|---|
| $||\boldsymbol{\nabla}_{\hat{y}}\mathcal{L}|| \propto$ | 1 | $1-\lambda$ | $1-2\lambda+\lambda^2$ | $1-3\lambda+3\lambda^2-\lambda^3$ | 1 | $(1-\lambda)$ | $(1-\lambda)^2$ | $(1-\lambda)^3$ |
| $||\epsilon_{L-1}|| \propto$ | 0 | $\lambda$ | $2\lambda-2\lambda^2$ | $3\lambda-6\lambda^2+3\lambda^3$ | 0 | $\lambda$ | $2\lambda(1-\lambda)$ | $3\lambda(1-\lambda)^2$ |
| $||\epsilon_{L-2}|| \propto$ | 0 | 0 | $\lambda^2$ | $3\lambda^2-3\lambda^3$ | 0 | 0 | $\lambda^2$ | $3\lambda^2(1-\lambda)$ |
| $||\epsilon_{L-3}|| \propto$ | 0 | 0 | 0 | $\lambda^3$ | 0 | 0 | 0 | $\lambda^3$ |

with $=$ between the two halves.

The state at time $t=0$ follows from feedforward state initialization, where all internal errors begin at zero. By construction, every entry in the table equals the sum of $(1-\lambda)$ times its left neighbor (its previous value) and $\lambda$ times its upper-left neighbor (influence from the layer above).

## B.3. A Binomial Formula for Signal Propagation

Examining the coefficient patterns reveals a fundamental mathematical structure: Pascal's triangle. We can formalize this behavior with the following binomial formula:

$$||\epsilon_{L-i}^t|| \propto \binom{t}{i} \lambda^i (1-\lambda)^{t-i}, \tag{6}$$

where $L + 1$ represents the total number of layers, $i$ is the distance from the output layer, and $t$ denotes the update step. The binomial coefficient $\binom{t}{i}$ encapsulates the number of possible paths through which a signal from the output layer can reach layer $L - i$ within exactly $t$ update steps, given our top-down propagation assumption.

Aside from the initial signal $\nabla_{\hat{y}} \mathcal{L}$ at the output, the formula reveals three additional factors that influence signal magnitude throughout the network:

1. **Exponential depth decay** $\lambda^i$: confirms the exponential attenuation with network depth identified in Section 3. This explains PC's exponential energy decay across layers, as first observed by Pinchetti et al. (2025) and later reproduced by Qi et al. (2025).

2. **Temporal decay** $(1 - \lambda)^{t-i}$: represents the gradual weakening of the original output signal over time, an artifact of our assumption that energy flows exclusively toward lower layers.

3. **Different propagation routes** $\binom{t}{i}$: accounts for the spatio-temporal variety of signal propagation pathways from output to the current layer.

### B.4. High-Precision sPC Simulations: Perfect Signal Propagation, Still Slow Convergence

The binomial formula of Eq. (6) serves as a powerful analytical tool to study sPC dynamics without the confounding effects of numerical precision limitations. By implementing this formula directly in logarithmic space using `scipy.special.gammaln`, we can achieve near-infinite precision and observe the theoretical behavior of signals in sPC unhindered by computational constraints.

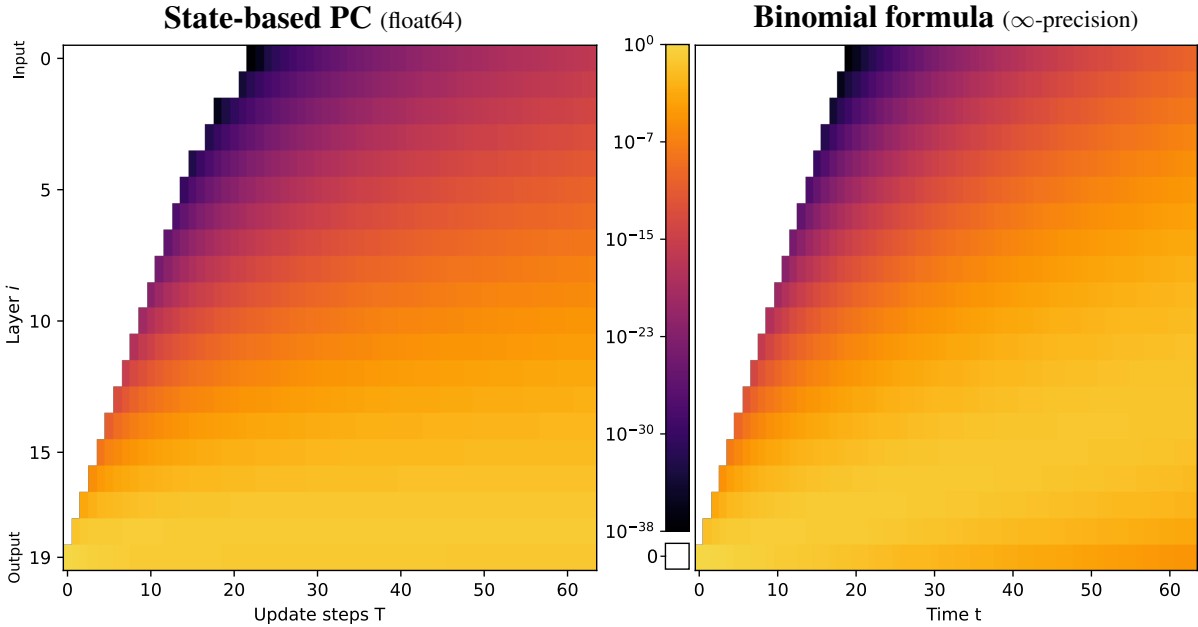

*Figure B.1.* Evolution of layerwise energies for sPC with float64 vs. near-infinite precision (simulated via Eq. (6)). Same setup as in Fig. 1, described in Appendix E.2.1.

Fig. B.1 presents a direct comparison between our high-precision binomial model and a float64 implementation of sPC for $\lambda = 0.1$. The striking similarity between these plots confirms that our mathematical characterization accurately captures the fundamental early-stage dynamics, despite simplifying assumptions. The orthogonal weight initialization used in our experiments certainly helps here, as it aligns well with our simplified backward dynamics assumption.

Comparing the float64 implementation in Fig. B.1 with the float32 version from Fig. 1 highlights both the importance and limitations of numerical precision in sPC. Even with enhanced double-precision floating-point arithmetic, the discontinuous signal propagation persists, though manifesting later and less pronounced.

*Table B.1.* Properties of a physical implementation vs. digital simulation of PC

| Property | Physics-based analog system | Digital simulation |
|---|---|---|
| Time dynamics | Continuous-time | Discrete-time |
| Time constant | $\tau \to 0$ | $0 < \lambda < 1$ (large for speed, small for stability) |
| Iterations per second | Near-infinite $(= 1/\tau)$ | A few dozen on consumer hardware |
| Power consumption | Very low | Relatively high |
| Compute budget | Scales with time | Scales with energy cost |
| Generality | Specialised (computes only itself) | General-purpose |

## B.5. The Impact of Signal Decay in Digital Simulation vs. in Physical Reality

### B.5.1. DIFFERENT HARDWARE, DIFFERENT PROPERTIES

The defining feature of sPC, and the cause of its popularity, is that it can be implemented in a physical substrate, like an analog circuit. But given sPC's scaling issues, is it still interesting to develop such hardware? Yes, we believe it is. The crucial consideration lies in the unique properties of a physical implementation, compared to a digital simulation. We have listed a comparison in Table B.1.

While signal decay remains a fundamental, hardware-independent property of sPC, it forms only a transient effect and does not necessarily pose a problem. The reason it becomes problematic in a digital simulation, is because the compute budgets are typically limited in practice, meaning that the (simulated) system does not receive sufficient iterations to reach convergence. By contrast, a physical chip could easily run millions of iterations per second, at neglegible monetary cost thanks to its low power consumption. The trade-off is that the chip can only compute this specific PC system, unlike a general-purpose digital processor.

Therefore, we may conclude that neuromorphic implementations of sPC, like the brain, would not suffer from the exponential signal decay problem identified in our research (although other challenges will emerge, such as noise and non-idealities).

### B.5.2. OUR SIMPLIFIED SPC MODEL, IN CONTINUOUS-TIME

To get some intuition for how signal propagation would look like in a physical substrate, we examine our binomial formula (Eq. (6)) in the continuous-time limit, where $\lambda \to 0$ (infinitesimal state learning rate) and $t \to \infty$ (continuous updates), with a constant total time $\lambda t$.

Setting $t = \tau/\lambda$, we find:

$$
\lim_{\lambda \to 0} ||\boldsymbol{\epsilon}_{\boldsymbol{L-i}}^{t=\tau/\lambda}|| \propto \lim_{\lambda \to 0} \binom{\tau/\lambda}{i} \lambda^i (1-\lambda)^{\tau/\lambda-i}
$$

$$
\approx \lim_{\lambda \to 0} \underbrace{\frac{(\tau/\lambda)^i}{i!}}_{\text{(Stirling's approximation)}} \lambda^i \underbrace{e^{-\tau}}_{\text{(limit definition of } e)}
$$

$$
= \frac{\tau^i}{i!} e^{-\tau}
$$

Now, our binomial formula transforms into a Poisson distribution, representing the spatial profile of a diffusion process. In this regime, signals diffuse smoothly over time, rather than being subject to the stepwise attenuation seen in discrete updates. Although there is still some signal decay present across layers (governed by the factorial $i!$), keep in mind that this formula only applies to our simplified model, not to sPC in general.

## B.6. Temporal Scope and Limitations

Our binomial model primarily captures early-stage dynamics but becomes progressively less accurate for extended optimization periods. For longer time horizons, especially with larger learning rates, our assumption of fixed predictions becomes increasingly unrealistic. The model dictates permanent downward energy transmission, moving from output to input, due to a lack of bottom-up signals. In practical implementations, however, layerwise energies will settle across the network in an

effort to minimize prediction errors globally. In particular, the output loss $\mathcal{L}$ will generally remain relatively large, even at equilibrium.

## C. Error-based PC and Connections to Other Algorithms

This section examines how ePC related to other established learning algorithms. More specifically, we show that:

- Despite its lack of locality, ePC is still a valid PC algorithm, according to the definition by Salvatori et al. (2026). (Appendix C.1)

- ePC and sPC are essentially the same PC algorithm, but follow different optimization paths. (Appendix C.2)

- Under specific conditions, ePC can implement exact backpropagation, closely resembling *Zero-divergent Inference Learning* (Z-IL; Song et al., 2020). (Appendix C.3)

- In general, however, ePC generates weight gradients that are different from those in standard backpropagation, matching those produced by full-equilibrium sPC. (Appendix C.4)

- When considering PC as a hierarchical probabilistic model, ePC simply implements the VAE reparametrization trick from Kingma and Welling (2013). (Appendix C.5)

### C.1. ePC Broadens the Definition of Predictive Coding

This appendix section analyzes the technical definition of PC by Salvatori et al. (2026) and demonstrates that ePC meets all criteria, thereby definitively establishing it as an exact PC algorithm. Moreover, we show that ePC broadens our understanding of what PC is, eliminating the condition of local-only interactions, commonly believed to be a core component of PC.

---

***Informal definition of PC by Salvatori et al. (2026), adapted to our notation.*** *Let us assume that we have a hierarchical generative model $g(\boldsymbol{x}, \boldsymbol{s})$, inverted using an algorithm $\mathcal{A}$.*
*Then, $\mathcal{A}$ is a Predictive Coding algorithm if and only if:*

1. *it maximizes the model evidence $\log p(\boldsymbol{s})$ by minimizing a variational free energy,*

2. *the posterior distributions of the nodes of the hierarchical structure are factorized via a mean-field approximation, and*

3. *each posterior distribution is approximated under the Laplace approximation (i.e., random effects are Gaussian).*

*Note that the above definition does not say anything explicitly about prediction error or properties such as locality, which, as mentioned earlier, are commonly used to describe PC. [...] the above definition is quite general and does not impose any constraint on the exact computation of the posteriors as well as the optimization technique(s) used to minimize the variational free energy.*

---

Let us go over each of the requirements of the definition:

0. ePC employs the exact same hierarchical generative model $g(\boldsymbol{x}, \boldsymbol{s})$ as sPC, as reflected by its identical energy function (see Appendix C.2). A reparametrization does not affect this property.

1. ePC minimizes sPC's variational free energy $E$ and reaches the same state minima (see Appendix C.2). As in sPC, this energy minimum corresponds to a maximum-likelihood estimation of the model evidence over the states.

2. ePC imposes a mean-field approximation of the posterior distribution. Concretely, this means that every state component $\boldsymbol{s}_{ij}$ can be set independently of any other state component. Although ePC builds a global computational graph, thereby imposing a dependence of $\boldsymbol{s}_i$ on all previous states $\boldsymbol{s}_{<i}$, we can still set $\boldsymbol{s}_i$ to any arbitrary value by modifying $\boldsymbol{\epsilon}_i$.

3. ePC's 'random effects' are the errors $\boldsymbol{\epsilon}$, which are implicitly modelled as Gaussians. More specifically, the energy $E$ (representing $-\log g(\boldsymbol{x}, \boldsymbol{s})$) contains a $\|\boldsymbol{\epsilon}_i\|^2$ term that corresponds to the negative log-likelihood of a standard Gaussian $\mathcal{N}(\mathbf{0}, \mathbf{1})$. This is entirely analogous to the reparametrization trick in VAEs (Kingma & Welling, 2013), which formulates any Gaussian as a transformation of a standard normal. We explore this connection further in Appendix C.5.

Remarkably, requirement 2 does *not* imply the need for locality, despite Salvatori et al. (2026) noting that *"the mean field approximation enforces independence, and hence, results in locality in the update rules"*. ePC proves that a non-local mean-field approximation exists, without violating the assumptions of the hierarchical model.

Technically speaking, one might argue that the final node in our model, $\hat{\boldsymbol{y}}$, is not independent of the other nodes. However, the loss $\mathcal{L}$ still provides the necessary freedom to deviate from $\boldsymbol{y}$, effectively acting as an additional random effect. Along with the definition's assumption of Gaussian posterior distributions, the loss $\mathcal{L}$ becomes an MSE loss and may be equivalently modelled as an additional error term $\boldsymbol{\epsilon}_L$.

### C.2. Theoretical Equivalence between sPC and ePC

Here, we provide a formal proof of the theoretical equivalence between the state- and error-based formulations of PC. We demonstrate that despite their different parameterizations, both approaches converge to identical equilibrium points and represent the same underlying optimization problem.

**Bijective Mapping**    We first establish a bijective mapping between the optimization variables of sPC (the states $\boldsymbol{s}$) and ePC (the errors $\boldsymbol{\epsilon}$).

**Theorem C.1** (Bijective Mapping). *For any fixed set of parameters $\boldsymbol{\theta}$ and input $\boldsymbol{x}$, there exists a bijective mapping between any state configuration $\boldsymbol{s} = \{\boldsymbol{s_0}, \boldsymbol{s_1}, \ldots, \boldsymbol{s_{L-1}}\}$ in sPC and error configuration $\boldsymbol{\epsilon} = \{\boldsymbol{\epsilon_0}, \boldsymbol{\epsilon_1}, \ldots, \boldsymbol{\epsilon_{L-1}}\}$ in ePC.*

*Proof.* Given states $\boldsymbol{s} = \{\boldsymbol{s_0}, \boldsymbol{s_1}, \ldots, \boldsymbol{s_{L-1}}\}$ in sPC, we can directly compute the corresponding errors:

$$\boldsymbol{\epsilon_i} = \boldsymbol{s_i} - \hat{\boldsymbol{s}}_i = \boldsymbol{s_i} - \boldsymbol{f_{\theta_i}}(\boldsymbol{s_{i-1}}) \quad \text{for} \quad i \in \{0, 1, \ldots, L-1\}$$

where for $i = 0$, we define $\boldsymbol{s_{-1}} := \boldsymbol{x}$ (the input data).

Conversely, given errors $\boldsymbol{\epsilon} = \{\boldsymbol{\epsilon_0}, \boldsymbol{\epsilon_1}, \ldots, \boldsymbol{\epsilon_{L-1}}\}$ in ePC and input $\boldsymbol{x}$, we can recursively compute the corresponding states:

$$\boldsymbol{s_0} = \hat{\boldsymbol{s}}_0 + \boldsymbol{\epsilon_0} = \boldsymbol{f_{\theta_0}}(\boldsymbol{x}) + \boldsymbol{\epsilon_0}$$
$$\boldsymbol{s_i} = \hat{\boldsymbol{s}}_i + \boldsymbol{\epsilon_i} = \boldsymbol{f_{\theta_i}}(\boldsymbol{s_{i-1}}) + \boldsymbol{\epsilon_i} \quad \text{for} \quad i \in \{1, 2, \ldots, L-1\}$$

For a fixed set of parameters $\boldsymbol{\theta}$ and input $\boldsymbol{x}$, this mapping is one-to-one and onto (i.e., bijective): for any given $\boldsymbol{s}$, there is exactly one corresponding $\boldsymbol{\epsilon}$, and for any given $\boldsymbol{\epsilon}$, there is exactly one corresponding $\boldsymbol{s}$. $\square$

**Energy Function Equivalence**    Next, we prove that under this mapping, the energy functions of both formulations are equivalent.

**Theorem C.2** (Energy Equivalence). *Under the bijective mapping between $\boldsymbol{s}$ and $\boldsymbol{\epsilon}$, for any fixed parameter set $\boldsymbol{\theta}$ and input-output pair $(\boldsymbol{x}, \boldsymbol{y})$, the energy functions $E_{sPC}(\boldsymbol{s}, \boldsymbol{\theta})$ and $E_{ePC}(\boldsymbol{\epsilon}, \boldsymbol{\theta})$ are identical when evaluated on corresponding configurations.*

*Proof.* Let us first recall the energy functions for both formulations:

$$E_{\text{sPC}}(\boldsymbol{s}, \boldsymbol{\theta}) = \frac{1}{2} \sum_{i=0}^{L-1} \|\boldsymbol{s_i} - \hat{\boldsymbol{s}}_i\|^2 + \mathcal{L}(\hat{\boldsymbol{y}}, \boldsymbol{y})$$

$$E_{\text{ePC}}(\boldsymbol{\epsilon}, \boldsymbol{\theta}) = \frac{1}{2} \sum_{i=0}^{L-1} \|\boldsymbol{\epsilon_i}\|^2 + \mathcal{L}(\hat{\boldsymbol{y}}, \boldsymbol{y})$$

Starting with $E_{\text{ePC}}$ and substituting the definition $\epsilon_i = s_i - \hat{s}_i$:

$$E_{\text{ePC}}(\epsilon, \theta) = \frac{1}{2} \sum_{i=0}^{L-1} \|\epsilon_i\|^2 + \mathcal{L}(\hat{y}, y)$$

$$= \frac{1}{2} \sum_{i=0}^{L-1} \|s_i - \hat{s}_i\|^2 + \mathcal{L}(\hat{y}, y)$$

$$= E_{\text{sPC}}(s, \theta)$$

Therefore, the energy functions evaluate to the same value for corresponding configurations of states and errors. $\qquad\square$

**Jacobian of the Transformation**   To analyze how gradients and critical points relate between the two formulations, we need the Jacobian matrix of the transformation from errors to states.

**Lemma C.3** (Jacobian Structure). *The Jacobian matrix $J = \frac{\partial s}{\partial \epsilon}$ representing how states change with respect to errors has a lower triangular structure with identity matrices on the diagonal.*

*Proof.* From the recursive definition of states in terms of errors:

$$s_i = f_{\theta_i}(s_{i-1}) + \epsilon_i$$

Taking partial derivatives with respect to $\epsilon_j$:

1. If $j > i$: $\frac{\partial s_i}{\partial \epsilon_j} = 0$, since $s_i$ doesn't depend on future errors.

2. If $j = i$: $\frac{\partial s_i}{\partial \epsilon_i} = I$, the identity matrix.

3. If $j < i$: $\frac{\partial s_i}{\partial \epsilon_j} = \frac{\partial f_{\theta_i}(s_{i-1})}{\partial s_{i-1}} \cdot \frac{\partial s_{i-1}}{\partial \epsilon_j}$

Let's denote $J_i = \frac{\partial f_{\theta_{i+1}}(s_i)}{\partial s_i}$ as the Jacobian of layer $i+1$ with respect to its input (state $i$).

Then we can write:

$$\frac{\partial s_i}{\partial \epsilon_j} = \begin{cases} 0 & \text{if } j > i \\ I & \text{if } j = i \\ J_{i-1} \cdot \frac{\partial s_{i-1}}{\partial \epsilon_j} & \text{if } j < i \end{cases}$$

This recursive structure leads to a lower triangular Jacobian matrix with identity matrices on the diagonal:

$$J = \begin{bmatrix} I & 0 & 0 & \cdots & 0 \\ J_0 & I & 0 & \cdots & 0 \\ J_1 J_0 & J_1 & I & \cdots & 0 \\ \vdots & \vdots & \vdots & \ddots & \vdots \\ \prod_{k=0}^{L-2} J_k & \prod_{k=1}^{L-2} J_k & \cdots & J_{L-2} & I \end{bmatrix}$$

This structure has important implications: $J$ is invertible with determinant 1, since the determinant of a triangular matrix is the product of its diagonal entries, all of which are 1. $\qquad\square$

**Gradient Equivalence and Critical Points**   We now establish the relationship between gradients in both formulations and use it to prove that they share the same critical points.

**Theorem C.4** (Gradient Relationship). *The gradients of the energy functions in the sPC and ePC formulations are related by:*

$$\nabla_{\boldsymbol{\epsilon}} E_{ePC} = \mathbf{J}^T \nabla_{\boldsymbol{s}} E_{sPC}$$

*where* $\mathbf{J} = \frac{\partial \boldsymbol{s}}{\partial \boldsymbol{\epsilon}}$ *is the Jacobian matrix derived in Lemma C.3.*

*Proof.* By the chain rule of calculus:

$$\nabla_{\boldsymbol{\epsilon}} E_{\text{ePC}} = \nabla_{\boldsymbol{\epsilon}} E_{\text{sPC}}(\boldsymbol{s}(\boldsymbol{\epsilon}), \boldsymbol{\theta})$$
$$= \left( \frac{\partial \boldsymbol{s}}{\partial \boldsymbol{\epsilon}} \right)^T \nabla_{\boldsymbol{s}} E_{\text{sPC}}$$
$$= \mathbf{J}^T \nabla_{\boldsymbol{s}} E_{\text{sPC}}$$

$\square$

**Theorem C.5** (Critical Point Correspondence). *A configuration* $\boldsymbol{s}^*$ *is a critical point of* $E_{sPC}$ *if and only if the corresponding configuration* $\boldsymbol{\epsilon}^*$ *is a critical point of* $E_{ePC}$.

*Proof.* From Theorem C.4, we have:

$$\nabla_{\boldsymbol{\epsilon}} E_{\text{ePC}} = \mathbf{J}^T \nabla_{\boldsymbol{s}} E_{\text{sPC}}$$

Since $\mathbf{J}$ is invertible (as shown in Lemma C.3), its transpose $\mathbf{J}^T$ is also invertible. Therefore:

$$\nabla_{\boldsymbol{\epsilon}} E_{\text{ePC}} = \mathbf{0} \iff \mathbf{J}^T \nabla_{\boldsymbol{s}} E_{\text{sPC}} = \mathbf{0} \iff \nabla_{\boldsymbol{s}} E_{\text{sPC}} = \mathbf{0}$$

This establishes that $\boldsymbol{s}^*$ is a critical point of $E_{\text{sPC}}$ if and only if the corresponding $\boldsymbol{\epsilon}^*$ is a critical point of $E_{\text{ePC}}$. $\square$

**Local Structure of Critical Points**  To complete our proof of optimization equivalence, we need to show that the local structure of critical points (minima, maxima, or saddle points) is preserved between formulations.

**Theorem C.6** (Preservation of Local Structure). *A critical point* $\boldsymbol{s}^*$ *is a local minimum / maximum / saddle point of* $E_{sPC}$ *if and only if the corresponding critical point* $\boldsymbol{\epsilon}^*$ *is a local minimum / maximum / saddle point of* $E_{ePC}$.

*Proof.* The local structure of critical points is determined by the eigenvalues of the Hessian matrices:

$$\mathbf{H}_{\boldsymbol{s}} = \nabla_{\boldsymbol{s}}^2 E_{\text{sPC}}$$
$$\mathbf{H}_{\boldsymbol{\epsilon}} = \nabla_{\boldsymbol{\epsilon}}^2 E_{\text{ePC}}$$

To relate these Hessians, we differentiate the relationship in Theorem C.4:

$$\nabla_{\boldsymbol{\epsilon}} E_{\text{ePC}} = \mathbf{J}^T \nabla_{\boldsymbol{s}} E_{\text{sPC}}$$

Taking another derivative with respect to $\boldsymbol{\epsilon}$:

$$\nabla_{\boldsymbol{\epsilon}}^2 E_{\text{ePC}} = \frac{\partial}{\partial \boldsymbol{\epsilon}} \left( \mathbf{J}^T \nabla_{\boldsymbol{s}} E_{\text{sPC}} \right)$$
$$= \frac{\partial \mathbf{J}^T}{\partial \boldsymbol{\epsilon}} \nabla_{\boldsymbol{s}} E_{\text{sPC}} + \mathbf{J}^T \frac{\partial \nabla_{\boldsymbol{s}} E_{\text{sPC}}}{\partial \boldsymbol{\epsilon}}$$
$$= \frac{\partial \mathbf{J}^T}{\partial \boldsymbol{\epsilon}} \nabla_{\boldsymbol{s}} E_{\text{sPC}} + \mathbf{J}^T \nabla_{\boldsymbol{s}}^2 E_{\text{sPC}} \frac{\partial \boldsymbol{s}}{\partial \boldsymbol{\epsilon}}$$
$$= \frac{\partial \mathbf{J}^T}{\partial \boldsymbol{\epsilon}} \nabla_{\boldsymbol{s}} E_{\text{sPC}} + \mathbf{J}^T \mathbf{H}_{\boldsymbol{s}} \mathbf{J}$$

At a critical point where $\nabla_s E_{\text{sPC}} = \mathbf{0}$, the first term vanishes, giving:

$$\mathbf{H}_\epsilon = \mathbf{J}^T \mathbf{H}_s \mathbf{J}$$

This establishes that $\mathbf{H}_\epsilon$ and $\mathbf{H}_s$ are congruent matrices, considering $\mathbf{J}$ is invertible.

By Sylvester's law of inertia, congruent matrices have the same number of positive, negative, and zero eigenvalues. Therefore:

- $\mathbf{H}_s$ is positive definite (all eigenvalues positive) if and only if $\mathbf{H}_\epsilon$ is positive definite

- $\mathbf{H}_s$ is negative definite (all eigenvalues negative) if and only if $\mathbf{H}_\epsilon$ is negative definite

- $\mathbf{H}_s$ has mixed positive/negative eigenvalues (saddle point) if and only if $\mathbf{H}_\epsilon$ has the same eigenvalue signature

This preserves the classification of critical points as local minima, maxima, or saddle points between the two formulations.

$\square$

**Dynamical Systems Analysis**    While the energy functions and their critical points are identical, the optimization dynamics differ significantly due to the reparameterization.

**Theorem C.7** (Dynamical Equivalence). *The continuous-time dynamics in sPC and ePC both converge to the same equilibrium points, but follow different trajectories in their respective spaces.*

*Proof.* Under the notation $\dot{x} := \frac{dx}{dt}$, the continuous-time dynamics for both formulations are:

$$\text{sPC:} \quad \dot{s} = -\nabla_s E_{\text{sPC}}$$
$$\text{ePC:} \quad \dot{\epsilon} = -\nabla_\epsilon E_{\text{ePC}}$$

Using the relation $\nabla_\epsilon E_{\text{ePC}} = \mathbf{J}^T \nabla_s E_{\text{sPC}}$, the ePC dynamics can be rewritten as:

$$\dot{\epsilon} = -\mathbf{J}^T \nabla_s E_{\text{sPC}}$$

To compare these dynamics in the same space, we need to transform $\dot{\epsilon}$ to $\dot{s}$. Using the chain rule:

$$\dot{s} = \frac{\partial s}{\partial \epsilon} \dot{\epsilon}$$
$$= \mathbf{J} \dot{\epsilon}$$
$$= -\mathbf{J}\mathbf{J}^T \nabla_s E_{\text{sPC}}$$

Comparing with the sPC dynamics:

$$\dot{s}_{sPC} = -\nabla_s E_{\text{sPC}}$$
$$\dot{s}_{ePC} = -\mathbf{J}\mathbf{J}^T \nabla_s E_{\text{sPC}}$$

The difference is the matrix $\mathbf{J}\mathbf{J}^T$, which acts as a preconditioner for the gradient descent. This matrix is positive definite (since $\mathbf{J}$ has full rank), meaning that the ePC dynamics will always move in a descent direction for $E_{\text{sPC}}$, but with a different step size and direction than sPC.

Both dynamical systems will converge to the same equilibrium points where $\nabla_s E_{\text{sPC}} = \mathbf{0}$, but will follow different trajectories to get there. Whereas sPC suffers from an ill-conditioned optimization landscape (Innocenti et al., 2025), explaining its slow convergence, ePC seems to solve this problem through a cleverly constructed preconditioner.    $\square$

**Global Connectivity and Signal Propagation**    The key computational advantage of ePC over sPC lies in its global connectivity structure. This difference affects how signals propagate through the network.

**Theorem C.8** (Signal Propagation). *In the sPC formulation, signals propagate sequentially through the network layers, resulting in exponential decay with network depth. In contrast, ePC allows direct signal propagation to all layers simultaneously, eliminating the signal decay problem.*

*Proof.* In sPC, the state update equations are:

$$\dot{\boldsymbol{s}}_{\boldsymbol{i}} = -\boldsymbol{\nabla}_{\boldsymbol{s}_{\boldsymbol{i}}} E_{\text{sPC}}$$
$$= -\boldsymbol{\epsilon}_{\boldsymbol{i}} + \left( \frac{\partial \boldsymbol{f}_{\boldsymbol{\theta}_{i+1}}}{\partial \boldsymbol{s}_{\boldsymbol{i}}} \right)^T \boldsymbol{\epsilon}_{\boldsymbol{i+1}}$$

The crucial observation is that $\dot{\boldsymbol{s}}_{\boldsymbol{i}}$ depends only on errors from adjacent layers ($\boldsymbol{\epsilon}_{\boldsymbol{i}}$ and $\boldsymbol{\epsilon}_{\boldsymbol{i+1}}$). This local connectivity means that a signal from the output layer ($\boldsymbol{\nabla}_{\hat{\boldsymbol{y}}}\mathcal{L}$) must propagate through all intermediate layers to reach the input layer, attenuating at each step.

In ePC, by contrast, the gradient is computed through the entire computational graph:

$$\dot{\boldsymbol{\epsilon}}_{\boldsymbol{i}} = -\boldsymbol{\nabla}_{\boldsymbol{\epsilon}_{\boldsymbol{i}}} E_{\text{ePC}}$$
$$= -\boldsymbol{\epsilon}_{\boldsymbol{i}} - \frac{\partial \hat{\boldsymbol{y}}}{\partial \boldsymbol{\epsilon}_{\boldsymbol{i}}}^T \boldsymbol{\nabla}_{\hat{\boldsymbol{y}}}\mathcal{L}$$

$$\text{with} \qquad \frac{\partial \hat{\boldsymbol{y}}}{\partial \boldsymbol{\epsilon}_{\boldsymbol{i}}} = \frac{\partial \hat{\boldsymbol{y}}}{\partial \boldsymbol{s}_{\boldsymbol{L-1}}} \cdot \frac{\partial \boldsymbol{s}_{\boldsymbol{L-1}}}{\partial \boldsymbol{s}_{\boldsymbol{L-2}}} \cdot \ldots \cdot \frac{\partial \boldsymbol{s}_{\boldsymbol{i}}}{\partial \boldsymbol{\epsilon}_{\boldsymbol{i}}}$$

Hence, $\dot{\boldsymbol{\epsilon}}_{\boldsymbol{i}}$ directly depends on all states from layer $i$ to $L$. This global connectivity allows signals from the output layer to immediately affect all earlier layers, eliminating the signal decay problem.

The mathematical consequence of this difference is that in sPC, the influence of the output error on layer $i$ decreases exponentially with the distance from the output (as explained extensively in Section 3), while in ePC, this influence is direct and unattenuated. $\qquad\square$

**Limitations and Caveats**    While the two formulations are theoretically equivalent in terms of equilibrium points, several practical considerations affect their performance:

1. **Optimization Landscape**: The different parameterizations create different state trajectories that may encounter different local minima under stochastic optimization.

2. **Numerical Stability**: The formulations may exhibit different numerical properties, particularly with respect to hyperparameter sensitivity and discretization effects. For instance, in Section 3, the numerical issues of sPC are highlighted.

3. **Implementation Efficiency**: The global connectivity of ePC imposes different computational demands than the local connectivity of sPC, affecting implementation efficiency on different hardware architectures. On GPU, the reverse-mode AD underlying ePC is highly efficient, despite being sequential. However, the local and parallel nature of sPC enables a far more efficient neuromorphic implementation.

Despite these practical differences, our theoretical equivalence analysis confirms that ePC is a valid reparameterization of PC that preserves its fundamental principles while offering significant computational advantages for deep networks.

## C.3. Exact Backpropagation using Error-based Predictive Coding

Below, we explore an important theoretical property of ePC: under specific conditions, ePC can become mathematically equivalent to standard backpropagation. This relationship deserves careful examination, as it affects how researchers should implement and interpret ePC results.

Note that, in general and under more reasonable circumstances, ePC does *not* equal backpropagation, and produces notably different weight gradients, as demonstrated in Appendix C.4.

### C.3.1. WHEN ePC REDUCES TO BACKPROPAGATION

The use of reverse-mode AD within ePC's computational structure naturally raises the question of when the two methods become mathematically equivalent. We demonstrate that ePC may reduce to standard backpropagation under specific conditions. Importantly, these conditions do *not* make backprop a PC algorithm, which would require, at least in theory, for the errors to be at equilibrium.

> **Theorem C.9.** *ePC becomes mathematically equivalent to backpropagation when either:*
>
>   - *The number of update steps $T$ is exactly 1.*
>   - *The error learning rate $\lambda$ is sufficiently small relative to $1/T$.*

*Proof.* We consider each case separately, proving their equivalence to backpropagation.

**Case 1: Single Update Step ($T = 1$)**
With a single update step, the error variables are updated to:

$$\epsilon_i = -\lambda \nabla_{\epsilon_i} \mathcal{L}(\hat{y}, y)$$

The subsequent parameter update becomes:

$$\Delta \theta_i \propto -\frac{\partial \hat{s}_i}{\partial \theta_i}^T \epsilon_i = \lambda \frac{\partial \hat{s}_i}{\partial \theta_i}^T \nabla_{\epsilon_i} \mathcal{L}(\hat{y}, y)$$

$$= \lambda \frac{\partial \hat{s}_i}{\partial \theta_i}^T \underbrace{\frac{\partial s_i}{\partial \hat{s}_i}^T}_{\equiv I} \underbrace{\cancel{\frac{\partial s_i}{\partial \epsilon_i}}^T}_{\equiv I} \nabla_{s_i} \mathcal{L}(\hat{y}, y)$$

$$= \lambda \nabla_{\theta_i} \mathcal{L}(\hat{y}, y)$$

This is precisely the gradient from standard backpropagation, but scaled by the error learning rate $\lambda$. Note that the weight update itself would involve an additional scaling by the *weight* learning rate.

When $\lambda = 1$, we find that this setup exactly matches that of *Zero-divergent Inference Learning* (Z-IL; Song et al., 2020). Z-IL adds a "fixed prediction assumption" to sPC and only models a backward signal wavefront, similar to that of Section 3, but with $\lambda = 1$. With these two constraints added to sPC, Z-IL effectively implements a 1-step version of ePC, which, as we outlined above, indeed corresponds to exact backpropagation.

**Case 2: Small Learning Rate ($\lambda \ll 1/T$)**
For a small $\lambda$, after $T$ update steps, the error can be approximated as a linear accumulation of $T$ identical updates:

$$\epsilon_i \approx -\lambda T \nabla_{\epsilon_i} \mathcal{L}(\hat{y}, y)$$

Following the same reasoning as for case 1, the resulting parameter update now becomes:

$$\Delta \theta_i \propto \lambda T \nabla_{\theta_i} \mathcal{L}(\hat{y}, y)$$

This approximation holds when $\lambda T$ remains sufficiently small, such that the error-perturbed output prediction $\hat{y}$ still closely approximates the unperturbed feedforward prediction $\hat{y}$ of backprop. Note that, when using larger learning rates and/or sufficient update steps, this is no longer the case, and the output prediction $\hat{y}$ differs sufficiently between ePC and backprop, with the natural consequence being notably distinct gradients. $\square$

### C.3.2. EXPERIMENTAL CONSIDERATIONS

In our experiments, we found that smaller values of $\lambda T$ generally performed best. However, we deliberately maintained this value above the threshold that would cause ePC to reduce to regular backpropagation. Complete experimental details are provided in Appendix E.

### C.3.3. CONTRAST WITH SPC

This situation differs notably from sPC, which can also become equivalent to backpropagation under certain conditions (Song et al., 2020; Millidge et al., 2022b). However, these conditions typically involve specific algorithmic tweaks that rarely occur in practice, thereby protecting sPC implementations from accidentally reducing to backpropagation.

ePC, by contrast, presents a more subtle boundary. During hyperparameter tuning, one might inadvertently select learning rates and iteration counts that effectively transform ePC into standard backpropagation. Researchers working with ePC should therefore carefully monitor these parameters to ensure they are truly studying PC dynamics rather than rediscovering backprop in disguise.

### C.4. Weight Gradients from (Error-Based) PC are Distinct from Backprop

In Section 4.3, we analyzed the evolution of states to equilibrium for a 20-layer linear PC network trained on MNIST. Here, we examine the evolution of the weight gradients themselves, which are ultimately the quantities of interest for learning.

While weight gradients in PC have no inherent dynamics (they are computed only after energy minimization), we can track how they would evolve if optimization were stopped at intermediate steps using their local formulas (Eq. (3)). The results are shown in Fig. C.1.

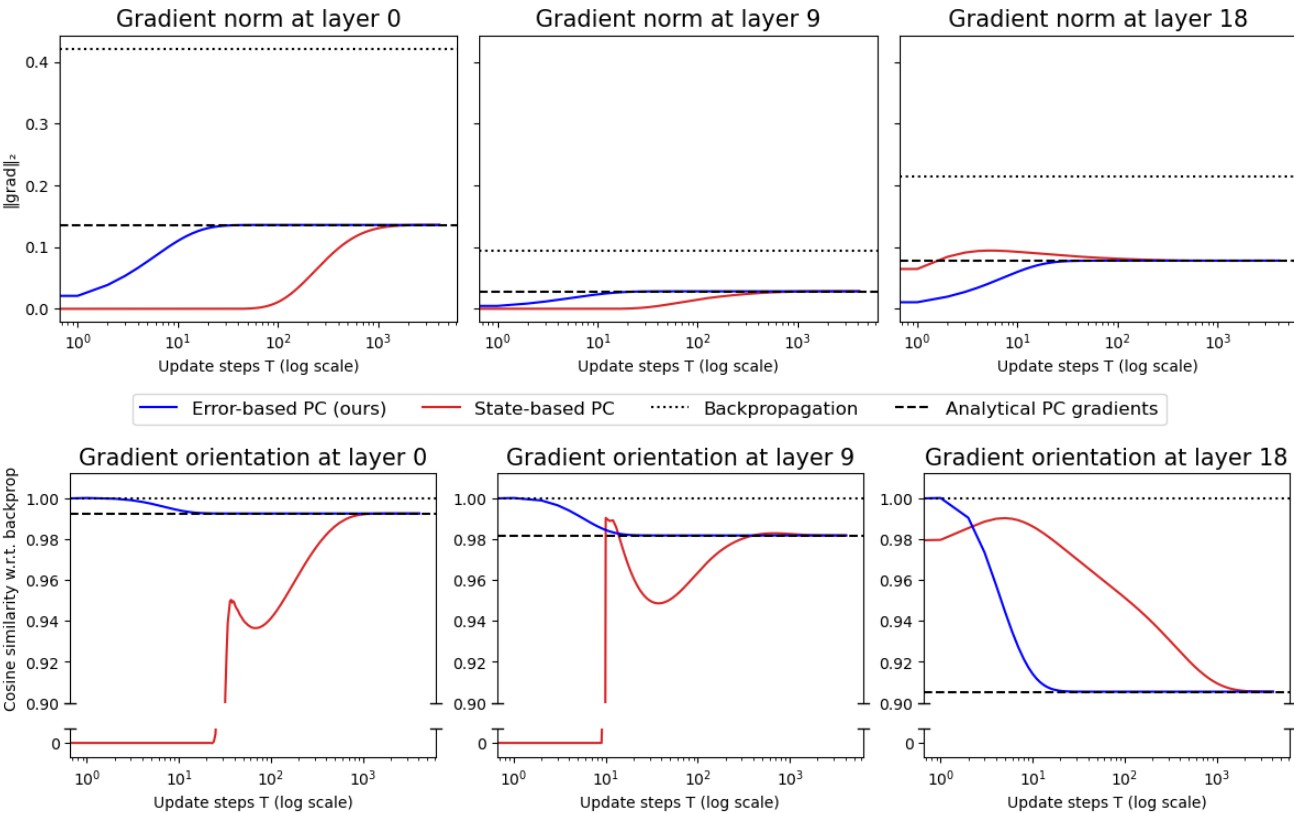

*Figure C.1.* Evolution of batch-averaged weight gradients of bottom, middle, and top hidden layers in a 20-layer linear PC network trained on MNIST (same setup as Fig. 5). ePC starts at backprop (rescaled by the error learning rate $\lambda$), while sPC starts mostly at zero. Both eventually reach the analytical PC gradients (notably distinct from backprop), with ePC converging roughly $100\times$ faster than sPC.

The analysis reveals several key insights about the gradient behavior of the two PC variants. Both ePC and sPC eventually converge to identical analytical PC weight gradients, reconfirming their theoretical equivalence from Appendix C.2. Notably, these PC gradients are distinct from backpropagation gradients across all layers, both in direction and in magnitude.

---

**How does ePC differ from backprop?**
While backpropagation and ePC do share the same computational backbone, they
ultimately solve very different optimization problems. A brief comparison:

| Solver methods: | BP | ePC |
|---|---|---|
| **Optimization problem** | Minimize loss $\mathcal{L}$ | Minimize PC energy $E$ |
| **Primary variables** | Model weights $\theta$ | PC errors $\epsilon$ |
| **Method structure** | One forward-backward pass | Iterative until convergence |
| **Computational backbone** | Reverse-mode AD | Reverse-mode AD |

---

When comparing ePC to sPC, the gradient dynamics differ dramatically. At $T = 1$, ePC starts at $\lambda$-rescaled backprop gradients (as shown in Appendix C.3) and rapidly transitions toward the PC solution. By contrast, sPC appears to transition from exact-zero gradients directly to PC.[3]

As expected, sPC suffers from severe signal propagation issues. Deeper layers stay at zero weight gradients for a long time, while layer 18 (the only initially non-zero gradient) actually becomes *more* aligned with backprop before slowly moving toward PC. This creates a significant risk, where early (pre-equilibrium) termination of sPC surely implements something other than true PC, despite achieving reasonable learning performance with extremely small yet informative gradients.

### C.5. ePC implements the VAE reparametrization trick in PC

Although PC has long been described as a variational Bayes algorithm (Friston & Kiebel, 2009; Bogacz, 2017; Millidge et al., 2021), it is only recently that its probabilistic generative properties have been explored (Oliviers et al., 2024; Zahid et al., 2024). In this context, PC is seen as a hierarchical Gaussian graphical model, where:

- Layer predictions $\hat{s}_i$ represent the predicted means ($\mu$) of Gaussian distributions (where negative log-likelihood corresponds to a squared error loss)

- Variance is typically fixed at $\sigma = 1$ (or precision weights are introduced)

- The states $s$ are sampled from the predicted Gaussians: $s_i \sim \mathcal{N}(\hat{s}_i, 1)$

- Standard PC's "energy minimization" (as described in Section 2) corresponds to finding the maximum-likelihood states $s$ instead of sampling from the full distribution

When ported to this setting, ePC becomes: $\qquad s_i = \hat{s}_i + 1 \odot \epsilon_i \quad$ (where 1 represents unit variance)

A common problem in probabilistic graphical models is that direct sampling breaks the computational graph, inhibiting the gradient flow needed for reverse-mode AD. One ingenious and highly successful solution is the VAE reparameterization trick (Kingma & Welling, 2013), which transforms a standard normal into the predicted distribution:

$$z = \mu + \sigma \odot \epsilon, \quad \epsilon \sim \mathcal{N}(0, 1)$$

Notice the strong similarity with the earlier formulation of ePC when $z \to s, \quad \mu \to \hat{s}_i, \quad \sigma \to 1$

Our description of ePC in Section 4.1 uses variational inference to find maximum-likelihood values for $\epsilon_i$ rather than drawing samples. However, under PC's probabilistic interpretation, one could sample the error variables $\epsilon_i$ from a standard normal distribution, making the connection to the VAE reparameterization trick more direct. The mathematical structure is entirely analogous: both methods reparameterize in terms of error/noise variables to enable efficient gradient flow.

---

[3]The directional wandering is likely just an artifact from the cosine similarity with a near-zero vector.

# D. Additional Experiments with Deep PC MLPs

This appendix contains additional experiments on deep non-linear networks. The results demonstrate that ePC's convergence advantages hold across non-linear architectures too, at least in our proof-of-concept setting.

## D.1. Proper Weight Initialization in sPC Helps To Mask Signal Decay, Not Solve It

### D.1.1. $\mu$PC ONLY TRAINS THE TOP PART OF THE NETWORK, LEAVING THE REST UNTRAINED

In response to the observation by Pinchetti et al. (2025) that sPC doesn't scale to deep architectures, Innocenti et al. (2025) suggested a clever weight initialization scheme, $\mu$PC, and demonstrated that it is sufficient to train a 128-layer sPC residual network, attaining excellent test accuracies using only 128 update steps. If signal decay is a fundamental limitation of sPC (as we claim), how can such a deep network be trained with so few steps? The answer: only a subset of the layers are actually trained.

To demonstrate this, we reproduced the setup of $\mu$PC and the 128-layer network of Innocenti et al. (2025). Using the same set-up as Fig. 1, we run a single training step on a freshly initialized $\mu$PC network and track the dynamics of the layerwise energies. The results are shown in Fig. D.1.

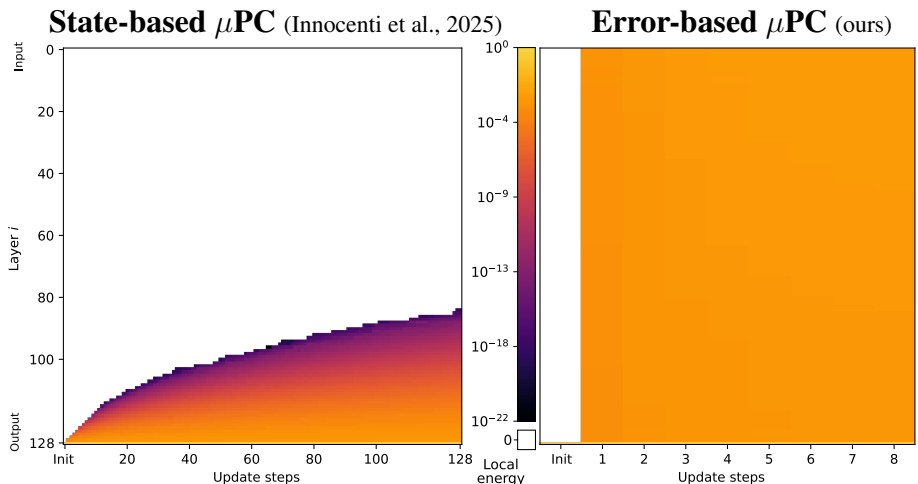

*Figure D.1.* Same setup as Fig. 1, but for the 128-layer $\mu$PC network of Innocenti et al. (2025). Again, sPC struggles to propagate the signal throughout the network, whereas ePC converges within a few steps.

As expected, **skip connections and careful weight initialization do not help to solve the signal decay problem.** Under an insufficient amount of update steps, the bottommost layers are left without gradient information, and thus generate exact-zero weight gradients under PC's local formulas of Eq. (3). As predicted in Section 3.3, this leads to a silent flaw in the training procedure, where a large amount of layers are left untrained. This phenomenon only gets worse throughout training, because the gradient signals will only get smaller as the model becomes better at predicting the output.

But if not all layers are trained, why does the model still attain such good test accuracies? This is where the beauty of $\mu$PC comes in. Any weight initialisation for deep networks must pass on the input signal with as little distortion as possible. This is a necessary (but not sufficient) condition for training deep networks, and explains why the addition of identity signal paths makes ResNets (He et al., 2016) so powerful. As such, the 80+ untrained layers of Fig. D.1 do not affect the performance of the model; they simply act as pass-through layers, perserving most of the data's information, akin to a fixed preprocessing pipeline. And so, $\mu$PC *does* enable training of a 128-layer network with sPC and limited update steps, but sacrifices representational power, reducing the model to 80+ harmless input transformations in front of a 40-layer trained neural network. By contrast, ePC trains all layers of the network and converges almost instantaneously.

### D.1.2. ORTHOGONAL WEIGHT INITIALIZATION

As an addition to the benchmark of Pinchetti et al. (2025), we evaluated the performance for a 20-layer deep MLP. As this architecture presents training challenges even with backprop, we used orthogonal weight initialization for enhanced stability (Hu et al., 2020). The results are stated in Table D.1. The exact hyperparameters used can be found in Appendix E

*Table D.1.* Additional test accuracies (in %) of ePC, sPC and backprop for a deep MLP architecture. Bold indicates best results within confidence intervals (mean ± 1 std. dev.; taken over 5 seeds).

| Loss $\mathcal{L}$ | *Mean Squared Error* | | | *Cross-Entropy* | | |
|---|---|---|---|---|---|---|
| Training algorithm | ePC | sPC | Backprop | ePC | sPC | Backprop |
| Deep MLP (20 layers) | | | | | | |
| MNIST | $97.11^{\pm 0.39}$ | $96.89^{\pm 0.33}$ | $\mathbf{97.89^{\pm 0.15}}$ | $94.84^{\pm 0.54}$ | $95.23^{\pm 1.24}$ | $\mathbf{97.20^{\pm 0.07}}$ |
| FashionMNIST | $\mathbf{85.04^{\pm 2.94}}$ | $84.92^{\pm 0.40}$ | $\mathbf{87.91^{\pm 0.45}}$ | $81.37^{\pm 0.40}$ | $79.95^{\pm 1.62}$ | $\mathbf{87.78^{\pm 0.18}}$ |

Despite our initial expectation of a large performance gap, both ePC and sPC performed similarly. The reason is the same as for $\mu$PC: untrained orthogonal layers do not degrade the input signal, and hence form a fixed, harmless preprocessing pipeline. As demonstrated in Fig. 1 (which uses the exact same architecture), the first few layers do not receive gradient information within the given update budget of 64 steps. Because the network is effectively reduced to a shallower subset, sPC generally underperforms ePC, even with this stable weight initialization.

To avoid confusion, we decided to put these results in the appendix, rather than in Section 5.

### D.2. State Dynamics in Deep Non-Linear Models Trained on MNIST

Building on our analysis of linear models in Section 4.3, we extend our investigation to the more practical scenario of non-linear networks. This extension allows us to evaluate whether the signal propagation advantages of ePC generalize beyond the analytically tractable linear case.

D.2.1. EXPERIMENTAL SETUP

We employed the 20-layer "Deep MLP" architecture detailed in Appendix E, pretrained on MNIST using one of two different loss functions: squared error and cross-entropy. Unlike the linear models, these non-linear networks achieve higher test accuracy (95% vs. 85%), representing a more realistic training scenario. However, this improved performance introduces an important methodological consideration: since analytical solutions are unavailable for non-linear models, we must turn to ePC's convergence state as our reference equilibrium point. This choice inherently favors ePC and should be considered when interpreting results.

D.2.2. IMPACT OF LOSS FUNCTION AND INPUT DIFFICULTY

As pointed out in Section 3, the signal decay problem of sPC depends critically on the output gradient $\nabla_{\hat{y}}\mathcal{L}$. In well-trained non-linear models, the loss—and hence its gradient—can become extremely small for easily classified examples, leading to even worse signal propagation issues. To investigate this effect systematically, we varied both the loss function (squared error vs. cross-entropy) and input difficulty (easy vs. hard-to-classify images). All experiments were implemented with float64 precision to ensure numerical stability and avoid precision-related confounders in the convergence analysis.

Fig. D.2 presents the convergence dynamics for both sPC and ePC across these conditions. Several important observations emerge from these experiments:

1. **Performance across loss functions:** Cross-entropy loss appears to create a more challenging optimization landscape for both sPC and ePC, despite its generally more favorable gradient properties compared to squared error. However, the smaller gradient signals from squared error do lead to long propagation delays in sPC, requiring almost 1000 update steps to progress just a single layer and reach $s_{18}$.

2. **Input difficulty effects:** For highly accurate models, most inputs will be "easy" (classified with high confidence), thereby generating minimal loss gradients. This greatly hinders overall signal propagation, even for ePC in our float64 simulations. By contrast, "hard" inputs (resulting in larger gradients) greatly accelerate ePC, while modestly improving sPC's convergence speed.

3. **Overall convergence speed:** ePC consistently converges orders of magnitude faster than sPC across all experimental conditions. This advantage is most pronounced with squared error loss on hard images (bottom left panel), where ePC converges approximately 10,000 times faster than sPC for the exact same model.

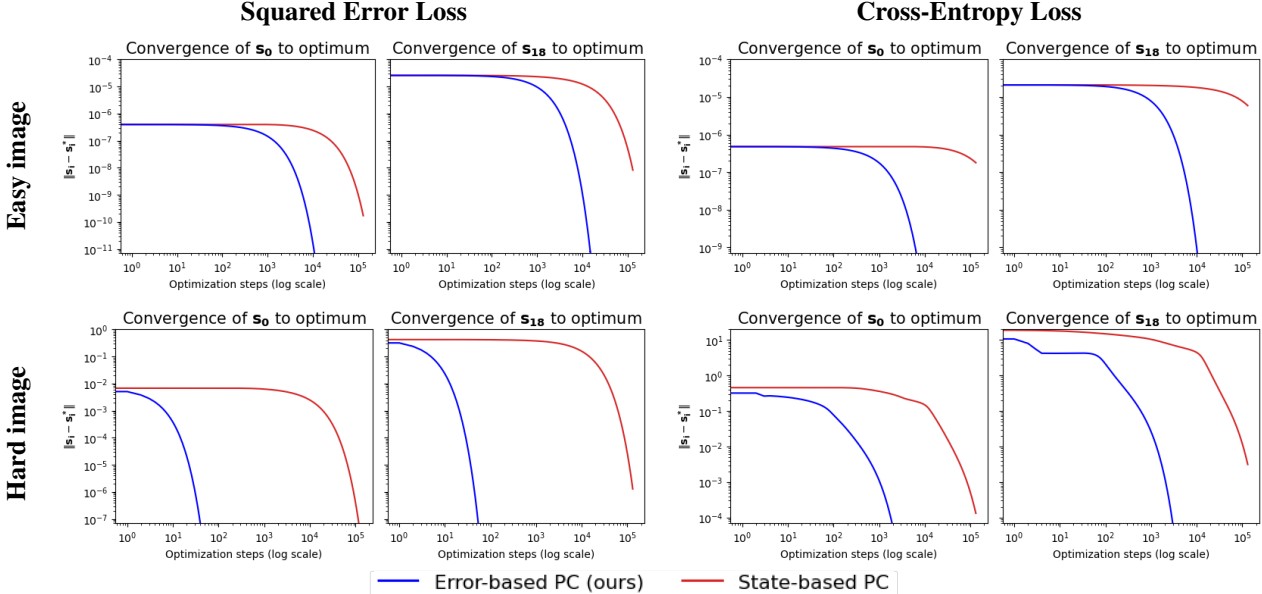

*Figure D.2.* Convergence dynamics in 20-layer PC-MLPs trained on MNIST with float64 precision. For easily-classified inputs (marked by near-zero loss), gradient signals become prohibitively small, hindering convergence. ePC consistently outperforms sPC by orders of magnitude across all conditions.

4. **Identical equilibria:** Even in the non-linear case, sPC and ePC seem to head towards the same equilibrium points, highlighting their equivalence as established in Appendix C.2. Note that this is not necessarily true for all model architectures and datasets, as spurious local minima may affect sPC and ePC in different ways.

5. **Practical implications:** Perhaps most critically, sPC requires an impractical number of update steps (>100,000) to reach convergence in these non-linear networks, underscoring the practical importance of our ePC reformulation for deep PC architectures.

These findings extend and reinforce the convergence analysis presented in Section 4.3. They confirm that the advantages of ePC's global connectivity structure generalize from the analytically tractable linear case to practical non-linear networks with different loss functions.

# E. Overview of Experimental Implementation Details

In this appendix, we provide all details necessary to reproduce our experimental results from Section 4.3 and Table 1. Our code is available at https://github.com/cgoemaere/error_based_PC.

## E.1. Details of Deep Linear Network Trained on MNIST

Below, we briefly summarize the technical details needed to reproduce Fig. 5. Specifically, we used the following architecture:

- **Number of layers**: 20
    - Specifically: $x - s_0 - s_1 - \cdots - s_8 - s_9 - s_{10} - \cdots - s_{17} - s_{18} - y$
      where '$-$' represents a layer (20 layers in total, leading to 19 hidden states $s_i$)

- **Hidden state dim**: 128

- **Activation function**: None (not even at the output)

- **Weight init**: orthogonal (linear gain) (Hu et al., 2020)

- **Bias init**: zero

- **State/error optimizer**: SGD

- **Pretraining**

  - **Weight optimizer**: Adam (Kingma & Ba, 2014)
  - **Weight learning rate**: 0.001 (not tuned for this proof-of-concept)
  - **Gradient algorithm**: Backpropagation (fast, stable, and neutral w.r.t. sPC & ePC)
  - **Dataset**: EMNIST-MNIST (Cohen et al., 2017)
  - **Batch size**: 64
  - **Epochs**: 2
  - **Final test accuracy**: 84.5%

For a fair comparison between sPC and ePC, we tuned the internal learning rate for both, with the objective of maximum convergence to the analytical optimum:

- **ePC**

  - **e_lr sweep**: {0.001, 0.005, 0.01, 0.05, 0.1}
  - **Optimal e_lr**: 0.05
  - **#iters**: 256

- **sPC**

  - **s_lr sweep**: {0.05, 0.1, 0.3, 0.5}
  - **Optimal s_lr**: 0.3
  - **#iters**: 4096

The analytical solution was obtained via sparse matrix inversion using `scipy.sparse.linalg.spsolve`.

**Note**: Fig. 5 shows *state* dynamics for both sPC and ePC. To get states for the latter, we project from errors to states at every time step.

### E.2. MLPs on MNIST & FashionMNIST

**Compute resources**

- **CPU**: Intel Xeon E5-2620 v4

- **RAM**: 32 GiB

- **GPU**: NVIDIA GeForce GTX 1080 Ti

- **Compute time per experiment**: *(without early stopping or failure)*

  - **MNIST - MLP**:
    * **ePC**: 10min (#iters=4, 25 epochs) – 1h (#iters=256, 5 epochs)
    * **sPC**: 10min (#iters=4, 25 epochs) – 45min (#iters=256, 5 epochs)
    * **Backprop**: 2-7min
  - **MNIST - Deep MLP**:
    * **ePC**: 12min (#iters=4, 25 epochs) – 3h (#iters=256, 5 epochs)
    * **sPC**: 25min (#iters=4, 25 epochs) – 3h (#iters=256, 5 epochs)
    * **Backprop**: 2-8min
  - **FashionMNIST - MLP**:
    * **ePC**: 6min (#iters=4, 14 epochs) – 14min (#iters=64, 5 epochs)

* * **sPC**: 13min (#iters=16, 14 epochs) – 45min (#iters=256, 5 epochs)
  * * **Backprop**: 3-7min
  - **FashionMNIST - Deep MLP**:
    * * **ePC**: 20min (#iters=4, 17 epochs) – 45min (#iters=64, 5 epochs)
    * * **sPC**: 1h (#iters=64, 7 epochs) – 5h30 (#iters=256, 13 epochs)
    * * **Backprop**: 4-7min

* **Total compute time estimate**:
  - **MNIST**: $\pm$150h
  - **FashionMNIST**: $\pm$150h

## Architecture

* **Number of layers**: 4 (MLP), 20 (Deep MLP; see below)

* **Hidden state dim**: 128

* **Activation function**: GELU (Hendrycks & Gimpel, 2016) (+ Sigmoid for MSE loss)

* **Weight init**: orthogonal (with ReLU gain) (Hu et al., 2020)

* **Bias init**: zero

* **Pseudorandom seed**: 42 for hyperparameter sweep, {0, 1, 2, 3, 4} for final test accuracy over 5 seeds. We set the seed using `lightning.seed_everything(workers=True)` before any data or weight initialization.

* **State/error optimizer**: SGD

* **Weight optimizer**: Adam (Kingma & Ba, 2014)

### E.2.1. DETAILS OF FIGURE 1

Below, we provide all details necessary to reproduce Fig. 1. Above all, our goal was to illustrate the problem of signal decay under realistic conditions.

The model is an untrained Deep MLP + Cross-Entropy, as detailed above. We track the layerwise energies throughout energy minimization for a single MNIST data pair $(\boldsymbol{x}, \boldsymbol{y})$. These are calculated using the energy functions corresponding to sPC and ePC (shown side-by-side in Fig. 2). We perform 64 update steps for sPC and 8 for ePC, both with a learning rate $\lambda = 0.1$.

### E.2.2. MNIST

**Data**  We used EMNIST-MNIST (Cohen et al., 2017), which is a well-documented reproduction of the original MNIST dataset (LeCun, 1998). The images are first rescaled to the range [0, 1], then they are normalized using the fixed values mean=0.5 and std=0.5 (same as Pinchetti et al., 2025). We set the batch size constant at 64. The validation set was 10% of the training data, split randomly but with a fixed seed. For final test performance, we don't split a separate validation set, but simply train on the whole training set.

**ePC**  First, we did a hyperparameter sweep over the inner optimization hyperparameters (error learning rate (e_lr) and number of update steps (#iters)), with the weight learning rate (w_lr) constant at 3e-4 (or 3e-5 for Deep MLP+CE). During these sweeps, we train for 5 epochs. Then, we fixed the best inner optimization hyperparameters for each setting, and tuned w_lr and the number of epochs by means of early stopping, with a maximum of 25 epochs. See Table E.1 for more details.

*Table E.1.* Hyperparameter sweep intervals and optimal values for ePC-MLPs on MNIST

| Hyperparams | Sweep values | MLP+MSE | MLP+CE | Deep MLP+MSE | Deep MLP+CE |
|---|---|---|---|---|---|
| e_lr | {0.001, 0.005, 0.01, 0.05, 0.1, 0.5} | 0.05 | 0.001 | 0.001 | 0.001 |
| #iters | {4, 16, 64, 256} | 4 | 4 | 4 | 4 |
| w_lr | {1e-5, 3e-5, 5e-5, 1e-4, 3e-4, 5e-4, 1e-3} | 1e-4 | 1e-4 | 1e-4 | 1e-5 |
| #epochs | Early Stopping(patience=3), up to 25 | 25 | 20 | 14 | 25 |

**sPC**    First, we did a hyperparameter sweep over the inner optimization hyperparameters (state learning rate (s_lr) and number of update steps (#iters)), with the weight learning rate (w_lr) constant at 1e-4 (the optimal rate for ePC). During these sweeps, we train for 5 epochs. Then, we fixed the best inner optimization hyperparameters for each setting, and tuned w_lr and the number of epochs by means of early stopping, with a maximum of 25 epochs. See Table E.2 for more details.

*Table E.2.* Hyperparameter sweep intervals and optimal values for sPC-MLPs on MNIST

| Hyperparams | Sweep values | MLP+MSE | MLP+CE | Deep MLP+MSE | Deep MLP+CE |
|---|---|---|---|---|---|
| s_lr | {0.01, 0.03, 0.1, 0.3} | 0.01 | 0.03 | 0.3 | 0.3 |
| #iters | {4, 16, 64, 256} | 16 | 4 | 64 | 64 |
| w_lr | {3e-5, 5e-5, 1e-4, 3e-4, 5e-4, 1e-3} | 1e-4 | 1e-4 | 1e-4 | 5e-5 |
| #epochs | Early Stopping(patience=3), up to 25 | 25 | 21 | 12 | 15 |

**Backprop**    Since there is no inner optimization in backprop, we simply tuned w_lr and the number of epochs by means of early stopping, with a maximum of 25 epochs. See Table E.3 for more details.

*Table E.3.* Hyperparameter sweep intervals and optimal values for backprop-MLPs on MNIST

| Hyperparams | Sweep values | MLP+MSE | MLP+CE | Deep MLP+MSE | Deep MLP+CE |
|---|---|---|---|---|---|
| w_lr | {3e-5, 5e-5, 1e-4, 3e-4, 5e-4, 1e-3} | 3e-4 | 1e-4 | 3e-4 | 5e-5 |
| #epochs | Early Stopping(patience=3), up to 25 | 16 | 20 | 22 | 18 |

### E.2.3. FASHIONMNIST

**Data**    We used the FashionMNIST dataset (Xiao et al., 2017). The images are first rescaled to the range [0, 1], then they are normalized using the fixed values mean=0.5 and std=0.5 (same as Pinchetti et al., 2025). We set the batch size constant at 64. The validation set was 10% of the training data, split randomly but with a fixed seed. For final test performance, we don't split a separate validation set, but simply train on the whole training set.

**ePC**    First, we did a hyperparameter sweep over the inner optimization hyperparameters (error learning rate (e_lr) and number of update steps (#iters)), with the weight learning rate (w_lr) constant at 1e-4 (3e-5 for Deep MLP+CE). During these sweeps, we train for 5 epochs. Then, we fixed the best inner optimization hyperparameters for each setting, and tuned w_lr and the number of epochs by means of early stopping, with a maximum of 25 epochs. See Table E.4 for more details.

*Table E.4.* Hyperparameter sweep intervals and optimal values for ePC-MLPs on FashionMNIST

| Hyperparams | Sweep values | MLP+MSE | MLP+CE | Deep MLP+MSE | Deep MLP+CE |
|---|---|---|---|---|---|
| e_lr | {0.001, 0.003, 0.01, 0.03, 0.1} | 0.01 | 0.003 | 0.001 | 0.001 |
| #iters | {4, 16, 64} | 16 | 4 | 4 | 4 |
| w_lr | {3e-5, 5e-5, 1e-4, 3e-4} | 5e-5 | 5e-5 | 3e-4 | 3e-5 |
| #epochs | Early Stopping(patience=3), up to 25 | 12 | 14 | 17 | 2 |

**sPC**   First, we did a hyperparameter sweep over the inner optimization hyperparameters (state learning rate (s_lr) and number of update steps (#iters)), with the weight learning rate (w_lr) constant at 1e-4. During these sweeps, we train for 5 epochs. Then, we fixed the best inner optimization hyperparameters for each setting, and tuned w_lr and the number of epochs by means of early stopping, with a maximum of 25 epochs. See Table E.5 for more details.

*Table E.5.* Hyperparameter sweep intervals and optimal values for sPC-MLPs on FashionMNIST

| Hyperparams | Sweep values | MLP+MSE | MLP+CE | Deep MLP+MSE | Deep MLP+CE |
|---|---|---|---|---|---|
| s_lr | {0.01, 0.03, 0.1, 0.3} | 0.03 | 0.01 | 0.3 | 0.1 |
| #iters | {4, 16, 64, 256} | 64 | 16 | 64 | 256 |
| w_lr | {3e-5, 5e-5, 1e-4, 3e-4} | 1e-4 | 1e-4 | 5e-5 | 3e-5 |
| #epochs | Early Stopping(patience=3), up to 25 | 14 | 14 | 7 | / |

**Backprop**   Since there is no inner optimization in backprop, we simply tuned w_lr and the number of epochs by means of early stopping, with a maximum of 25 epochs. See Table E.6 for more details.

*Table E.6.* Hyperparameter sweep intervals and optimal values for BP-MLPs on FashionMNIST

| Hyperparams | Sweep values | MLP+MSE | MLP+CE | Deep MLP+MSE | Deep MLP+CE |
|---|---|---|---|---|---|
| w_lr | {3e-5, 5e-5, 1e-4, 3e-4} | 3e-4 | 1e-4 | 1e-4 | 3e-4 |
| #epochs | Early Stopping(patience=3), up to 25 | 15 | 25 | 17 | 10 |

### E.3. VGG-models and ResNet-18

**Compute resources**   We report the resources used for training and each model's training time. The training times required for a model with MSE or CE loss are comparable. For hyperparameter tuning, we evaluate 200 distinct parameter configurations, making the total computational cost approximately 200 times greater than that of a single training run.

- **CPU**: Intel Xeon w5-3423

- **RAM**: 197 GiB

- **GPU**: NVIDIA RTX A6000

- **Compute time per experiment**: *(without early stopping or failure)*

| Dataset | Model | ePC | sPC | Backprop |
|---|---|---|---|---|
| CIFAR-10 | VGG-5 | 6min | 9min | 2min |
| | VGG-7 | 7min | 11min | 2min |
| | VGG-9 | 9min | 17min | 3min |
| | ResNet-18 | 29min | – | 6min |
| CIFAR-100 | VGG-5 | 6min | 9min | 2min |
| | VGG-7 | 7min | 12min | 2min |
| | VGG-9 | 9min | 19min | 3min |
| | ResNet-18 | 29min | – | 6min |

- **Total compute time estimate for tuning across model architecture and loss function**:
  - **ePC**: ±680h
  - **sPC**: ±510h
  - **Backprop**: ±170h

**Data** We used the CIFAR-10/100 datasets (Krizhevsky, 2009). The images are first rescaled to the range [0, 1], then they are normalized with the mean and standard deviation given in Table E.7 (same as Pinchetti et al., 2025). We set the batch size constant at 256. The validation set was 5% of the training data, split randomly but with a fixed seed. For final test performance, we don't split a separate validation set, but simply train on the whole training set.

*Table E.7.* Data normalization

|  | **Mean ($\mu$)** | **Std ($\sigma$)** |
|---|---|---|
| CIFAR-10 | [0.4914, 0.4822, 0.4465] | [0.2023, 0.1994, 0.2010] |
| CIFAR-100 | [0.5071, 0.4867, 0.4408] | [0.2675, 0.2565, 0.2761] |

**VGG architecture** VGG models are deep convolutional neural networks (Simonyan & Zisserman, 2014). Table E.8 provides a detailed summary of the model architectures used for the VGG-5, VGG-7 and VGG-9 models. After the convolutional layers, a single linear layer produces a class prediction. The activation function of the models was selected from among ReLU, Tanh, Leaky ReLU, and GELU (Hendrycks & Gimpel, 2016) during model tuning.

*Table E.8.* Detailed architectures of VGG models. The locations of the pooling layers correspond to the indices of the convolutional layers after which the max-pooling operations are applied.

|  | **VGG-5** | **VGG-7** | **VGG-9** |
|---|---|---|---|
| Channel Sizes | [128, 256, 512, 512] | [128, 128, 256, 256, 512, 512] | [128, 128, 256, 256, 512, 512, 512, 512] |
| Kernel Sizes | [3, 3, 3, 3] | [3, 3, 3, 3, 3, 3] | [3, 3, 3, 3, 3, 3, 3, 3] |
| Strides | [1, 1, 1, 1] | [1, 1, 1, 1, 1, 1] | [1, 1, 1, 1, 1, 1, 1, 1] |
| Paddings | [1, 1, 1, 1] | [1, 1, 1, 0, 1, 0] | [1, 1, 1, 1, 1, 1, 1, 1] |
| Pool location | [0, 1, 2, 3] | [0, 2, 4] | [0, 2, 4, 6] |
| Pool window | 2 × 2 | 2 × 2 | 2 × 2 |
| Pool stride | 2 | 2 | 2 |

**ResNet-18 architecture** The ResNet-18 model is a convolutional neural network with skip connections (He et al., 2016). Our implementation follows the standard ResNet-18 architecture with modifications tailored for CIFAR-10/100. It is composed of an initial convolutional stem followed by four residual stages, each consisting of two residual blocks. Each residual block comprises two 3×3 convolutional layers with batch normalization and ReLU activation, followed by an identity shortcut connection. Spatial downsampling is performed via stride-2 convolutions at the beginning of each stage beyond the first. Table E.9 details the layer configuration.

*Table E.9.* ResNet-18 architecture adapted for CIFAR-10/100 image classification. The feature shape describes the image height and width after each stage. The residual configuration gives the dimension of the convolution mask, the number of channels and the stride used for the residual stream. All the convolutional layers used a padding of one, and each convolution was followed by a batch normalisation layer. Stages one to four include skip connections for every residual.

| **Stage** | **Feature shape** | **Residual configuration** |
|---|---|---|
| Conv Stem | $32 \times 32$ | Conv3x3, 64, stride $= 1$ |
| Stage 1 | $32 \times 32$ | $\begin{bmatrix} \text{Conv3x3, 64, stride} = 1 \\ \text{Conv3x3, 64, stride} = 1 \end{bmatrix} \times 2$ |
| Stage 2 | $16 \times 16$ | $\begin{bmatrix} \text{Conv3x3, 128, stride} = 2 \\ \text{Conv3x3, 128, stride} = 1 \end{bmatrix} \times 2$ |
| Stage 3 | $8 \times 8$ | $\begin{bmatrix} \text{Conv3x3, 256, stride} = 2 \\ \text{Conv3x3, 256, stride} = 1 \end{bmatrix} \times 2$ |
| Stage 4 | $4 \times 4$ | $\begin{bmatrix} \text{Conv3x3, 512, stride} = 2 \\ \text{Conv3x3, 512, stride} = 1 \end{bmatrix} \times 2$ |
| Head | $1 \times 1$ | Global AvgPool + Linear classifier |

**Learning rate schedule**    The following learning rate schedule was used to help stabilize training:

1. For the first 10% of training, the learning rate increases linearly from w_lr up to $1.1 \times$ w_lr.

2. After the warmup phase, a cosine decay is applied. The learning rate smoothly decreases to $0.1 \times$ w_lr, following a cosine curve, for the remaining training steps.

**Weight initialization**    We used the default PyTorch weight initialization, which amounts to a random uniform weight and bias initialization. For pseudorandom seeds, we use 42 for the hyperparameter sweeps, and {0, 1, 2, 3, 42} for the final test accuracy over 5 seeds. We set the seed using `lightning.seed_everything(workers=True)` before any data or weight initialization.

*Table E.10.* Summary of hyperparameter tuning and training settings for convolutional models

| Method | Tuned hyperparameter range | Optimizer | Optim steps (T) | Epochs (sweep/final) |
|---|---|---|---|---|
| ePC | e_lr: fixed at 0.001
e_momentum: fixed at 0.0
w_lr: log-uniform [1e-5, 1e-2]
w_decay: log-uniform [1e-6, 1e-3] | SGD (error)
Adam (weights)
(Kingma & Ba, 2014) | 5 (all models) | 25/25 (VGG)
25/50 (ResNet-18) |
| sPC | s_lr: log-uniform [1e-3, 5e-1]
s_momentum: {0, 0.25, 0.5, 0.75, 0.9}
w_lr: log-uniform [1e-5, 1e-2]
w_decay: log-uniform [1e-6, 1e-3] | SGD (state)
AdamW (weights)
(Loshchilov & Hutter, 2019) | 8 (VGG-5)
10 (VGG-7)
12 (VGG-9) | 25/25 (VGG) |
| Backprop | w_lr: log-uniform [1e-5, 1e-2]
w_decay: log-uniform [1e-6, 1e-3] | Adam (weights) | — | 25/25 (VGG)
25/50 (ResNet-18) |

**Glossary:** w_lr: base weight learning rate (see learning rate schedule below), w_decay: weight decay, {e,s}_lr: error / state learning rate, {e,s}_momentum: error / state momentum, T: nr. of update steps

**Hyperparameter tuning**    We performed hyperparameter tuning using Hyperband Bayesian optimization provided by Weights and Biases. The search was conducted over the hyperparameter spaces specified in Table E.10 across different model architectures, datasets, and loss functions. All tuning was guided by top-1 validation accuracy as the primary objective. Final top-5 accuracy metrics reported in Table 1 are for the models that achieved the highest top-1 accuracy. The best hyperparameters for each model identified through the sweep are provided in Table E.11, as well as in the "configs_results/" folder of the codebase. The "configs_sweeps/" folder contains all the sweep configs.

*Table E.11.* Overview of optimal hyperparameter configurations, used in our experiments

| Data | Loss | Algo | Architecture | s/e_lr | s/e_momentum | w_lr | w_decay | act_fn |
|---|---|---|---|---|---|---|---|---|
| CIFAR-10 | Squared Error | sPC | VGG-5 | 2.66e-2 | 0 | 4.21e-4 | 2.68e-6 | gelu |
| | | | VGG-7 | 2.28e-3 | 0.05 | 2.07e-3 | 3.10e-6 | gelu |
| | | | VGG-9 | 1.73e-2 | 0.5 | 5.77e-5 | 6.49e-4 | tanh |
| | | ePC | VGG-5 | 0.001 | 0 | 4.71e-4 | 1.48e-5 | gelu |
| | | | VGG-7 | 0.001 | 0 | 4.26e-4 | 2.16e-6 | gelu |
| | | | VGG-9 | 0.001 | 0 | 6.61e-4 | 4.01e-5 | gelu |
| | | | ResNet-18 | 0.001 | 0 | 7.65e-4 | 1.82e-4 | — |
| | | BP | VGG-5 | — | — | 6.30e-4 | 1.09e-6 | gelu |
| | | | VGG-7 | — | — | 5.45e-4 | 1.37e-6 | gelu |
| | | | VGG-9 | — | — | 5.24e-4 | 1.27e-6 | gelu |
| | | | ResNet-18 | — | — | 3.00e-4 | 9.04e-4 | — |
| | Cross-Entropy | sPC | VGG-5 | 1.47e-2 | 0.05 | 2.64e-4 | 1.21e-5 | gelu |
| | | | VGG-7 | 1.59e-3 | 0 | 1.76e-3 | 1.03e-5 | gelu |
| | | | VGG-9 | 5.80e-2 | 0 | 8.09e-5 | 4.18e-5 | tanh |
| | | ePC | VGG-5 | 0.001 | 0 | 7.79e-4 | 1.72e-4 | gelu |
| | | | VGG-7 | 0.001 | 0 | 1.56e-3 | 5.46e-4 | gelu |
| | | | VGG-9 | 0.001 | 0 | 5.36e-4 | 6.88e-4 | tanh |
| | | | ResNet-18 | 0.001 | 0 | 3.39e-3 | 1.51e-6 | — |
| | | BP | VGG-5 | — | — | 1.66e-3 | 4.55e-4 | gelu |
| | | | VGG-7 | — | — | 1.10e-3 | 4.51e-5 | gelu |
| | | | VGG-9 | — | — | 6.21e-4 | 3.58e-5 | gelu |
| | | | ResNet-18 | — | — | 1.67e-3 | 1.49e-4 | — |
| CIFAR-100 | Squared Error | sPC | VGG-5 | 3.73e-3 | 0.75 | 9.80e-4 | 2.14e-6 | gelu |
| | | | VGG-7 | 1.44e-2 | 0 | 1.88e-4 | 9.38e-5 | tanh |
| | | | VGG-9 | 4.78e-2 | 0.25 | 7.07e-5 | 7.79e-5 | tanh |
| | | ePC | VGG-5 | 0.001 | 0 | 8.05e-4 | 2.33e-6 | gelu |
| | | | VGG-7 | 0.001 | 0 | 4.02e-4 | 1.47e-5 | gelu |
| | | | VGG-9 | 0.001 | 0 | 2.01e-4 | 2.62e-6 | gelu |
| | | | ResNet-18 | 0.001 | 0 | 3.67e-4 | 7.30e-4 | — |
| | | BP | VGG-5 | — | — | 4.57e-4 | 1.27e-5 | gelu |
| | | | VGG-7 | — | — | 4.47e-4 | 6.71e-6 | gelu |
| | | | VGG-9 | — | — | 4.70e-4 | 4.34e-6 | gelu |
| | | | ResNet-18 | — | — | 3.95e-4 | 5.45e-4 | — |
| | Cross-Entropy | sPC | VGG-5 | 2.13e-2 | 0 | 8.61e-4 | 1.48e-6 | tanh |
| | | | VGG-7 | 1.04e-1 | 0.5 | 3.00e-4 | 6.69e-5 | tanh |
| | | | VGG-9 | 1.25e-2 | 0.75 | 4.69e-4 | 3.45e-4 | tanh |
| | | ePC | VGG-5 | 0.001 | 0 | 8.27e-4 | 8.22e-4 | tanh |
| | | | VGG-7 | 0.001 | 0 | 3.13e-4 | 7.99e-4 | tanh |
| | | | VGG-9 | 0.001 | 0 | 3.23e-4 | 4.03e-4 | tanh |
| | | | ResNet-18 | 0.001 | 0 | 3.03e-3 | 1.20e-5 | — |
| | | BP | VGG-5 | — | — | 1.04e-3 | 7.69e-4 | gelu |
| | | | VGG-7 | — | — | 1.38e-3 | 4.13e-4 | gelu |
| | | | VGG-9 | — | — | 8.24e-4 | 1.62e-6 | gelu |
| | | | ResNet-18 | — | — | 1.33e-3 | 1.96e-4 | — |

