# OpenReview forum: "ePC: Fast and Deep Predictive Coding in Digital Simulation"
_ICML.cc/2026/Conference — ICML 2026 regular_

### Official Review · Reviewer_RS6e · 2026-02-23

**Soundness:** 2
**Presentation:** 4
**Significance:** 3
**Originality:** 2
**Overall Recommendation:** 3
**Confidence:** 4

**Summary:**

This paper identifies a fundamental limitation in standard Predictive Coding (PC) networks, referred to as state-based PC (sPC). The authors demonstrate that during energy minimization, error signals in sPC propagate through the network in a wave-like fashion, leading to exponential decay with depth—a problem that severely limits scalability in digital simulations. To address this, they propose error-based PC (ePC), a reparameterization that uses errors as the primary optimization variables. By restructuring the computation flow, ePC enables direct, unattenuated signal propagation via backpropagation. The paper provides theoretical analysis of the signal decay, proves the mathematical equivalence of sPC and ePC at equilibrium, and presents extensive experiments showing that ePC scales to deep architectures (VGG, ResNet) and achieves performance on par with backpropagation, whereas sPC fails.

**Compliance With Llm Reviewing Policy:**

Affirmed.

**Key Questions For Authors:**

1.  The paper acknowledges that without biological plausibility, ePC's main advantage is not yet clear. Beyond the scope of this work, what specific problem do you envision ePC being better suited for than standard backpropagation? Is it purely a tool for studying PC dynamics, or do you believe it has inherent algorithmic advantages (e.g., in continual learning, uncertainty estimation, or data efficiency) that future work will uncover?

2.  Your paper is pragmatically agnostic about biology, treating ePC as a tool for digital simulation and sPC as relevant for neuromorphic hardware. This creates two parallel research tracks. Do you see these tracks as converging or diverging? Will insights from ePC ultimately inform our understanding of biological learning, or will the two tracks remain separate, with ePC serving purely engineering goals and sPC serving purely neuroscientific ones? If they diverge, what is the intellectual value of maintaining the "PC" label across both tracks?

3.  Your paper solves the scaling problem by discarding locality and adopting backpropagation. This raises a fundamental question: In your view, what is the minimal necessary condition for an algorithm to be considered "Predictive Coding"? If an algorithm minimizes the PC energy function but does so via backprop (as ePC does), and another algorithm minimizes the same energy but does so via local, sequential updates (as sPC does), do they both represent the same underlying computational principle? In other words, is PC defined by its objective or by its mechanism?

**Limitations:**

Yes.

**Strengths And Weaknesses:**

Strengths

1.  Identifies a Fundamental Flaw: The paper provides a clear, novel, and mathematically rigorous diagnosis of why sPC fails to scale to deep networks (exponential signal decay). This is a significant contribution to the PC literature.

2.  Elegant and Effective Solution: The proposed ePC is an elegant reparameterization that directly solves the identified problem.

3.  Exceptional Clarity: The paper is exceptionally well-written. The figures are informative, the writing is crisp, and the argumentation is logical and easy to follow.


Weaknesses

1.  Abandonment of Biological Plausibility: The paper is upfront that ePC is not biologically plausible. This is a major concession, as biological plausibility has been the primary motivation for studying PC. By discarding this core feature, the paper cedes the very ground that made PC intellectually distinctive.

2.  Unclear Competitive Advantage Over Backpropagation: Without biological plausibility, ePC is presented as an algorithm that matches backpropagation. The paper does not adequately answer why one should adopt this more complex algorithm over standard, simpler, and equally performant backpropagation. The potential advantages hinted at (e.g., online/continual learning) are acknowledged but not explored.

3. Limited Experimental Scope: The experimental scope is limited to standard supervised image classification tasks. While sufficient to prove ePC's superiority over sPC, it does not explore domains where PC might have unique advantages, as the authors acknowledge in the limitations section. The hyperparameter tuning for the deep networks is extensive, which is good, but it also highlights the potential complexity of getting ePC to work, even if it's more stable than sPC.

---

> ### Author Rebuttal · Authors · 2026-03-30
>
> We thank the reviewer for their recognition of our work as "a significant contribution to the PC literature" and for their kind words on our manuscript's form and clarity.
>
> The reviewer asks about our definition for PC and our view on ePC's use cases.
>
> ‎
>
> > **When does an algorithm qualify as "Predictive Coding"?**
>
> We are very explicit about this: Appendix C.1 draws upon the definition of PC by Salvatori et al. (2026) to characterize ePC as an exact PC algorithm.
>
> Conceptually, PC is defined by its objective, namely the minimization of variational free energy. *How* this energy is minimized (i.e., the solver mechanism) is largely irrelevant; only the final equilibrium matters. sPC and ePC represent different solvers to the same PC objective.
>
> That's what makes PC unique, you have a free choice of solver: one is biologically plausible (but computationally intractable), the other is numerically efficient. This allows one to digitally simulate the outcome of a process implementable by biology, without being restricted to the physical limitations of nature. From that perspective, ePC extends rather than restricts PC's applicability.
>
> ‎
>
> > **What is ePC's main advantage over backprop? And will ePC evolve separately from sPC?**
>
> ePC's main advantage over backprop is the existence of an equivalent algorithm that can be directly mapped to neuromorphic hardware, which, in turn, promises to be fast and extremely energy-efficient.
>
> Notably, ePC does not preclude any property of sPC, because both methods are functionally interchangeable, modelling the same PC objective. One can use ePC for fast digital simulations of PC and transfer the conclusions to sPC. For hardware development, such digital simulations are absolutely crucial, and so we see the paths of "ePC for digital" and "sPC for neuromorphic" as strongly intertwined / converging. (We explain this view in more detail in our rebuttal to reviewer bXEP)
>
> We acknowledge that this "digital simulation for hardware" motivation is currently missing from the paper and will clarify this in the introduction.
>
> Apart from hardware, our "Conclusion and Future Directions" describes a second research track in the literature, that of PC's learning advantages over backprop.
>
> It's not clear how this track will evolve, as it depends on which specific mechanisms are revealed to provide benefits. Nonetheless, ePC's improved speed can help researchers investigate this question more thoroughly. In case the mechanisms would diverge from the PC definition, we agree that the "PC" label should be dropped.
>
> ‎
>
> > **Why not explore domains where PC might have an advantage over backprop?**
>
> We agree that such experiments would be interesting future avenues to apply ePC to. Importantly, as the reviewer notes, our current set of experiments is sufficient to support our main claims of ePC's superiority over sPC.
>
> As stated in our Section 5, our experimental scope was an intentional copy of the benchmarking setup of Pinchetti et al. (2025), specifically established to study the research question of why sPC doesn't scale. This allows for fair comparison with prior PC methods, which struggle to even match backprop's performance.
>
> The reviewer kindly recognizes our extensive hyperparameter tuning, which took considerable time. A rigorous study to validate PC's learning advantages (using ePC) would form an even more ambitious endeavor and deserves a dedicated paper of its own.
>
> ‎
>
> ---
>
> ‎
> ### **Conclusion**
>
> With the additional clarification of what constitutes a PC algorithm and how we see the future of ePC, we hope to have convinced the reviewer that our work is not only a significant contribution, but also worthy of acceptance. If so, we would be grateful if the reviewer would consider increasing their support for this paper.

---

> > ### Author Rebuttal · Reviewer_RS6e · 2026-04-03
> >
> > I appreciate the authors' clarifications. I will maintain my original decision, as this paper cannot adequately demonstrate the distinctive advantages of the ePC algorithm relative to existing neural networks.

---

> > > ### Author Response · Authors · 2026-04-03
> > >
> > > We are glad to hear all concerns have been adequately addressed. The final standpoint of the reviewer is surprising, because (as indicated in the first sentence of the abstract) **(e)PC is not a replacement for neural networks, but a learning algorithm for training them**.
> > >
> > > As for the advantages over backprop (the standard NN training algorithm), this has been clarified in the rebuttal. An empirical demonstration would require custom PC hardware that does not yet exist.
> > >
> > > The goal and claims of our paper are not to outperform backprop, but to establish a new performant PC algorithm that resolves a major bottleneck in PC research. In their original review, **the reviewer recognizes our success** in doing so, **describing it as "a significant contribution to PC literature"**.

---

### Official Review · Reviewer_rzwf · 2026-03-11

**Soundness:** 3
**Presentation:** 3
**Significance:** 2
**Originality:** 2
**Overall Recommendation:** 3
**Confidence:** 2

**Summary:**

This paper proposes error-based predictive coding (ePC), which addresses the inefficiencies and signal decay of state-based PC (sPC), enabling faster computation and scaling to deep architectures while matching backpropagation performance. However, ePC relies on mechanisms similar to backpropagation, sacrifices biological plausibility, and is validated on a limited set of datasets and architectures. Key questions include whether ePC loses sPC’s parallelization advantage, what practical benefits it offers over backpropagation, how it differs fundamentally from backpropagation, and whether it remains useful if efficient sPC hardware becomes available.

**Compliance With Llm Reviewing Policy:**

Affirmed.

**Key Questions For Authors:**

1.	Does the proposed ePC forfeit the potential advantage of parallelized training offered by sPC, given that ePC’s updates are performed sequentially?
2.	Compared to standard backpropagation, what specific advantages does ePC offer, e.g., faster convergence, reduced computational cost, or lower memory usage, considering that ePC incorporates backpropagation within its learning algorithm?
3.	In line with the previous point, what are the concrete differences between backpropagation and ePC, given that ePC effectively includes backpropagation in its update process?
4.	If future hardware were developed that could efficiently implement sPC, would ePC still provide practical or theoretical benefits over sPC in such a scenario?

**Limitations:**

yes

**Strengths And Weaknesses:**

Strength:

1.	This paper reformulates predictive coding (PC) to enable efficient implementation on digital hardware, addressing the inefficiencies of the canonical state-based PC (sPC).
2.	The proposed ePC eliminates the exponential signal decay that prevents sPC from effectively training deep neural networks.
3.	ePC achieves significantly faster computation, running orders of magnitude quicker than sPC.
4.	ePC can scale to deep architectures and consistently matches backpropagation performance across multiple datasets and network designs.

Weakness:

1.	The effectiveness of ePC appears to rely on mechanisms akin to global connectivity and backpropagation, which raises questions about the distinct technical contributions of this work compared to standard backpropagation methods.
2.	ePC seems to sacrifice biological plausibility to handle signal decay, which may have unintended consequences for the theoretical foundation and intrinsic mechanisms of predictive coding.
3.	The experimental evaluation is limited in scope, using only a small set of datasets and model architectures, which may not be sufficient to fully validate the generality and effectiveness of the proposed approach.
4.	The experimental use of a 20-layer deep linear network for MNIST classification is somewhat unrealistic, as such architectures are not commonly applied for this dataset.

---

> ### Author Rebuttal · Authors · 2026-03-30
>
> We thank the reviewer for their recognition of our proposed method and its effectiveness in solving PC's scaling problem efficiently.
>
> The reviewer raises several concerns about the nature of ePC and the consequences of our design choices.
>
> ‎
>
> > **How does ePC differ from backprop?**
>
> From an algorithmic perspective, backprop and ePC share the same backbone, but ultimately solve very different optimization problems. A brief comparison:
>
> | Solver methods: | BP | ePC
> |--|--|--
> | **Optimization problem** | Minimize loss $\mathcal{L}$ | Minimize PC energy $E$
> | **Primary variables** | Model weights $\mathbf{\theta}$ | PC errors $\mathbf{\epsilon}$
> | **Method structure** | One forward-backward pass | Iterative until convergence
> | **Computational backbone** | Reverse-mode automatic differentiation | Reverse-mode automatic differentiation
>
> Also behaviorally, the two methods produce different outcomes, as empirically established in Appendix C.4. In theory, a 1-step version of ePC would be exactly equivalent to backprop (as shown in Appendix C.3), but we don't consider this to be ePC.
>
> To help future readers, we will include this comparison in Appendix C.4.
>
> ‎
>
> > **What specific advantage does ePC offer over backprop? And does ePC affect PC's foundations?**
>
> We want to clarify the motivation for ePC, since this shapes the answer to both questions.
>
> From a Machine Learning perspective, PC is interesting for two reasons: 1) literature suggests PC can outperform backprop in specific settings, and 2) sPC allows for a neuromorphic implementation that is both fast and extremely energy-efficient. This is PC's most concrete advantage over backprop today, and forms the primary driver of our work. (We explain this in more detail in our reply to reviewer bXEP)
>
> But hardware development must start with digital simulations, and with sPC, this is not computationally feasible. A faster method is needed, but with a strong requirement: it must be an *exact* PC algorithm to maintain equivalence to sPC. This is where ePC comes in.
>
> In Appendix C.1, we formally establish ePC as an exact PC algorithm, using the PC definition by Salvatori et al. (2026). Preserving all foundations of PC, ePC is functionally interchangeable with sPC, which means conclusions drawn from ePC simulations transfer directly to sPC hardware.
>
> The one concession is biological plausibility, but we argue this is unavoidable. Signal decay is a direct consequence of sPC's local update rule, and no fix can preserve both biology and scalability. Also, note that we are already far from biological plausibility the moment we run anything on a digital computer.
>
> Ultimately, the goal of this work is not to compare ePC to backprop, but to sPC. This also explains our somewhat limited experimental scope, which directly mirrors the setup of Pinchetti et al. (2025), the community's accepted benchmark for comparing PC algorithms.
>
> We acknowledge the neuromorphic motivation is currently missing from the paper and will clarify this in the introduction.
>
> ‎
>
> > **Does ePC lose sPC's parallel advantage?**
>
> Yes, but ironically: the more one can parallelize sPC, the *greater* the benefit of switching to ePC.
>
> This is because sPC has two competing trends that scale with the number of layers ($L$):
> 1. **Parallelization advantage**: one iteration is ~ $\mathcal{O}(L)$ faster
> 2. **Signal decay problem**: the system requires ~$\mathcal{O}(\exp(L))$ more iterations to converge
>
> So, while parallelization can linearly speed-up a single iteration of sPC, the required number of iterations grows exponentially larger. As explained in our Clarifying Remarks (page 11), sPC *never* has a parallelization advantage over ePC.
>
> ‎
>
> > **If we had efficient sPC hardware, would ePC still have a purpose?**
>
> Absolutely!
>
> First off, we want to emphasize that ePC is an essential stepping stone towards developing this sPC hardware. Our rebuttal to reviewer bXEP discusses this in detail.
>
> Even if sPC hardware would exist, ePC still has the advantage of *flexibility*. An analog chip is rigid, a digital algorithm is not. We can use ePC to test out new ideas before porting them over to sPC hardware.
>
> ‎
>
> > **Why use a 20-layer linear network for MNIST?**
>
> It's true, this is a contrived architecture, specifically chosen to quantify ePC's superiority. To do so, we required:
> 1. **A deep network**: needed to prove that ePC succeeds where sPC struggles
> 2. **A linear network**: provides an analytical solution that can act as ground truth, allowing for a fair comparison
> 3. **A simple task**: because we are using a linear network
>
> We will add an explanation of this architectural choice to Section 4.3.
>
> ‎
>
> ---
>
> ‎
>
> We appreciate the reviewer's recognition of the effectiveness of ePC, and hope this rebuttal managed to position it as a fast exact-PC algorithm, notably different from backprop. With the nature of our method clarified, we would be grateful if the reviewer would consider increasing their support for this paper.

---

> > ### Author Rebuttal · Reviewer_rzwf · 2026-04-03
> >
> > Thank you for your efforts in improving the submission. Many of the concerns have been addressed through the author response, which is appreciated. However, a few points would still benefit from further clarification:
> >
> > * The performance of ePC seems to rely on mechanisms similar to global connectivity and backpropagation. It would be helpful if the authors could elaborate on this aspect in more detail.
> >
> > * Additional evaluations on a broader range of datasets would further strengthen the validation of the proposed method.

---

> > > ### Author Response · Authors · 2026-04-03
> > >
> > > We thank the reviewer for acknowledging our efforts and address the two points below. To respect the reviewer's time, we have kept the reply brief.
> > >
> > > Yes, ePC relies on global connectivity, and we want to be precise about what this implies:
> > > 1) As reviewer bXEP remarked, our findings suggest that global connectivity is required to make PC work in digital simulations. We prove that sPC's **locality is a fundamental flaw** that inhibits scaling (in digital simulations).
> > > 2) While ePC's solver *mechanism* resembles that of backprop, the solver itself and the problem it solves are both different. *(as explained in the rebuttal)*
> > > 3) In digital simulation, **global connectivity is actually *desirable***: GPUs are designed to handle exactly this, and ePC takes full advantage of it. That is why we explicitly describe ePC as "PC for digital", not "the best PC". It's all hardware-dependent.
> > >
> > > As for our experimental scope, it deliberately mirrors Pinchetti et al. (2025), an ICLR Spotlight paper that forms **the community's accepted benchmark for comparing PC algorithms**. Additional experiments could be interesting to further explore (e)PC's properties, but perhaps not strictly necessary to prove ePC's superiority over sPC (as acknowledged by reviewers Yknt and RS6e).
> > >
> > > Together with our rebuttal, we hope to have provided sufficient additional context to convince the reviewer of the merits of our work, and we remain available to answer any further questions.

---

### Official Review · Reviewer_bXEP · 2026-03-12

**Soundness:** 3
**Presentation:** 3
**Significance:** 2
**Originality:** 3
**Overall Recommendation:** 3
**Confidence:** 4

**Summary:**

This paper presents an improvement to predictive coding - a way to optimize ML models that is an alternative to back-propagation and more based on minimum energy principles.  It is inspired by neuromorphic systems that tend to self adapt to minimum energy states. Traditional PC seems to have problem with slow convergence, particularly in deep networks, and they present a new approach ePC that resolves this. The basic tradeoff is that the new approach requires more global updates - which is contrary to the neuromorphic systems they envision.

**Compliance With Llm Reviewing Policy:**

Affirmed.

**Key Questions For Authors:**

Can you quantify the cost of the ePC algorithm's global connectivity requirement?

Why do you say this is headed to digital hardware? Is this because the problem in ePC is related to "machine epsilon".  I would say that analog hardware would equally have this problem (because thermal noise would limit its ability to resolve small differences in values)?

**Limitations:**

Yes.

**Strengths And Weaknesses:**

The paper seems like a significant advance in the domain of predictive coding. They are getting much better results than PC on at least small benchmarks. The approach is well described and the results look sound.

However, it is completely unclear if will make a significant improvement over back-propagation.  Because they had to go towards more global updates, they are deviating away from the local connectivity and local updates typically desired in "neuromorphic systems".  There is a cost to this in terms of implementability and likely things like energy consumption which the paper dies not quantify.

In fact, it is unclear to this review how this is aimed towards digital hardware, as my understanding of digital neuromorphic systems (e.g., Loihi from Intel) is far more centered around implementing spiking neural networks (SNNs) for which back-prop is the desired training approach. Loihi does support some on-line training algorithms, but nothing like what I see needed in ePC. I personally think the desired behavior in PC and ePC is more likely a match to analog or next generation hardware (quantum or superconducting) designed to find minimum energy states.  I thus think the phrasing that this is headed towards "digital" hardware may be misleading.  That said, whether implemented digitally or otherwise, I think the global connectivity required in ePC is a detractor and if this is what is needed to make PC work for complex networks may explain why the entire approach is yet to be practical.

For these reasons, my largest concern is that it is absolutely unclear how much this moves predictive coding type algorithms towards practicality.

---

> ### Author Rebuttal · Authors · 2026-03-30
>
> We thank the reviewer for their enthusiasm for our work, describing it as "a significant advance in the domain of PC". We appreciate that the reviewer is already thinking ahead about the practical impact of our research, because this is exactly the right area to focus on.
>
> Please allow us to share our *concrete* research vision for PC.
>
> The most promising direction, which we are currently exploring, is **custom analog hardware for sPC**. Such hardware would enable fast and extremely energy-efficient AI training, providing a clear practical purpose for PC in the age of power-hungry hyperscalers. Yes, sPC may seem problematic *in simulation*, but that is only because digital compute is relatively slow and often limited in practice. By contrast, analog hardware "executes" a tremendous amount of time steps per second, reducing the signal decay problem to a barely perceptible transient effect. We touch upon this point in Appendix B.5.
>
> So how does ePC help towards this vision, and why focus on digital if analog is the end goal?
>
> **Analog chip development always begins with digital simulation.** In our case, we need to verify PC's robustness against various analog limitations, including device variability and thermal noise. **Testing this digitally is simply not viable with sPC**, for the reasons uncovered in this paper, and as the reviewer correctly remarks, global connectivity seems to be required to make PC simulations work. ePC allows us to simulate the necessary algorithmic tests and tweaks, bringing us one step closer to analog PC.
>
> We certainly did not mean to suggest a future of digital ePC chips. The end goal remains unchanged: analog sPC hardware. With ePC, we can finally take the first step in that direction, initiating digital simulations to verify PC behavior in an imperfect analog-like environment. We will add this framing of "digital as an essential first step towards analog" to the paper, and suggest to revise the paper's title from "for Digital Hardware" to "in Digital Simulation" to make the distinction clearer.
>
> ‎
>
> ### Key questions
> With the context above, let us address your two key questions directly:
> > What is the cost of the ePC's global connectivity?
>
> In digital simulation, there is no cost. On the contrary, it is an advantage, since GPUs are specifically designed to handle the global connectivity of backprop.
>
> In neuromorphic hardware, ePC is not applicable; it is intended for digital simulation only. But its algorithmic equivalence to sPC ensures it still plays an essential role in hardware development.
>
> > Why do you say PC is headed to digital hardware? And would analog also struggle with signal decay?
>
> We do not mean to suggest PC is headed to digital and will make the necessary adjustments to clarify this. Although PC started out digitally and got stuck there, we believe that, thanks to ePC, it will soon be headed to analog.
>
> The signal decay problem is a computational artifact, a practical consequence of the slowness of time passing in digital simulations. If we had faster digital compute, waiting for 100.000+ iterations (as required for sPC) would not be a problem, and sPC simulations would be viable. But this is not the case in practice.
>
> In an analog implementation of sPC, however, the actual physical time advances much faster than simulated time. In other words, analog "computes" significantly more sPC iterations per second than digital. Therefore, although sPC's mechanism of signal decay is still present, it no longer forms a practical hurdle.
>
> ‎
>
> ---
>
> ‎
> ### Conclusion
> Taken together, these clarifications position ePC as an enabler of a concrete hardware roadmap, rather than an end goal in itself. We hope this broader perspective demonstrates that our contribution holds clear practical impact beyond its algorithmic novelty. With the additional context, we would be grateful if the reviewer would consider increasing their support for this paper.

---

> > ### Author Rebuttal · Reviewer_bXEP · 2026-04-05
> >
> > I very much appreciate the authors clarification that they are headed towards analog hardware design. However, my concern of impact still remains. First, I think the authors should clarify that there are two types of analog hardware implementations - the first mimics digital, relying on analog multiply-accumulate and some form of nonlinearity (such as RELU) and ends with a Analog to Digital converter before implementing the final decision (e.g. arg max for classification).  This is in contrast to analog hardware that may emulate going to a minimum-energy state. This hardware is often limited by the physical nature of the chip - every neuron must have its own hardware which limits the scalability of such systems. Thus, I think the authors needs to be concrete about this approach as I am not sure which hardware they are talking about (as I am more familiar) with the more standard analog MAC/RELU approach.
> >
> > That all said, the biggest issue I see is that to get their approach to make sense they need more global connections and I think this will impact the implementability of even an analog hardware implementation.  Thus, while very interesting, I remain cautious as to whether or not this paper will have the impact it hopes to achieve.
> >
> > I suggest to make this paper acceptable for ICML that the authors be explicit about what analog hardware they are/can map to and ensure that for such hardware the more global connections they need are not a problem and to address any scalability issues that this hardware may have (that is, if my assumption that there is no reuse of hardware across multiple neurons/layers).

---

> > > ### Author Response · Authors · 2026-04-07
> > >
> > > Dear reviewer, thank you very much for your engagement.  We are happy to answer the final questions.
> > >
> > > **(1) Which analog hardware are we targeting?**
> > >
> > > We are targeting analog in-memory computing, based on resistive crossbar arrays (where synaptic weights are stored as the conductance values of non-volatile memristive devices, such as ReRAM/RRAM, PCM, or MRAM elements). This is the second class the reviewer describes: hardware that physically settles into energy-minimizing states.
> > > Such crossbar arrays naturally perform matrix vector products, as a physical relaxation to equilibrium (which corresponds to the state update phase in sPC); no clock cycles needed, since the hardware is the energy minimizer.
> > >
> > > We do not consider the analog MAC/ReLU pipeline the reviewer described first, where the analog part is used to accelerate the multiply-accumulate step of a digital algorithm, requiring an ADC at the output of the layers.  In the crossbar configuration we envision, signals remain analog throughout the network, and the equilibrium state of the entire circuit corresponds directly to the PC energy minimization.
> > >
> > > **(2) Does global connectivity in EO harm analog implementability?**
> > >
> > > EO's global connectivity is a property of our digital simulation, not of the target analog hardware. In EO, the errors are the optimization variables, and their gradient is computed via backpropagation: this is what requires global connectivity in simulation.
> > > However, the analog hardware target is sPC (state optimization), not ePC (error optimization). In sPC, all updates remain strictly local: each neuron only communicates with its immediate neighbors, and the energy minimization comes from the physical dynamics of the circuit itself. The global connectivity in EO is introduced specifically to make *digital simulation* of PC tractable. Once we move to analog hardware, sPC's local structure is restored and no neuron needs to communicate with distant layers.
> > >
> > > **(3) Potential scalability issues**
> > >
> > > Indeed, in the crossbar based approach a hardware neuron is not reused over multiple network positions: each synaptic weight requires its own memristive device.  However, this limitation is shared by all crossbar-based in memory compute approaches (whether trained with backpropagation or PC), and is the subject of active research (e.g. tiling strategies or 3D crossbar stacking [1]).  Our contribution does not worsen this constraint. On the contrary, by enabling reliable digital simulation of PC for the first time, EO is what makes it possible to *study* how PC behaves under the device variability and noise conditions specific to these crossbar arrays, which is an essential step before transitioning to analog.
> > >
> > >
> > > We hope this helps resolving the remaining doubts.  Thank you for your time and effort, and for the interesting points raised!
> > >
> > > [1] Choi, Bezugam, Bhattacharya, Kwon, Strukov, “Wafer-scale fabrication of memristive passive crossbar circuits for brain-scale neuromorphic computing”, Nature Communications 16(8757), 2025.

---

### Official Review · Reviewer_Yknt · 2026-03-13

**Soundness:** 4
**Presentation:** 4
**Significance:** 4
**Originality:** 3
**Overall Recommendation:** 5
**Confidence:** 4

**Summary:**

This paper addresses a limitation in state-based Predictive Coding (sPC): exponential signal decay during energy minimization when simulated on digital hardware. The authors of the paper explain that this decay mechanism explains both the slow convergence of sPC and its degradation in deep architectures. To solve this issue, they introduce error-based Predictive Coding (ePC), a technique that optimizes prediction errors rather than neural states, uses backpropagation internally to propagate signals and produces identical PC weight gradients at equilibrium.

**Compliance With Llm Reviewing Policy:**

Affirmed.

**Final Justification:**

The response was prepared well; overall, the paper is (was already) in good shape. My final recommendation is the acceptance.

**Key Questions For Authors:**

1- How is ePC fundamentally different from standard backpropagation at the algorithmic and implementation (e.g., hardware) levels, as it uses backpropagation during the error update step?

2- Can the authors provide more discussion or evidence regarding the digital hardware implications of ePC?

3- How well does ePC generalize beyond the evaluated standard image classification benchmarks to larger-scale or more diverse tasks?

**Limitations:**

yes

**Strengths And Weaknesses:**

The paper discusses the predictive coding (PC) concept as a brain-inspired alternative to backpropagation. The topic is timely, considering the physical system analogy.

PC's current digital simulation challenge is underlined as a motivation.

A well-prepared paper, from motivation to early proofs and the figures. Figures are very well prepared. Visual aids are used very professionally.

The proposed method is clear and sound. The step-by-step flow is given clearly in synchronisation with the math behind.

A very comprehensive and guiding Appendix was prepared.

Although the authors claim that ePC is still a Predictive Coding method because it produces the same weight updates as state-based PC at equilibrium, it actually uses backpropagation to compute gradients during the error update step. This may be a concern: if ePC relies on global backpropagation to send signals through the network, then from an algorithmic perspective, it may look very similar to standard backpropagation. Even though the final weight updates follow the PC formulation, the use of backprop makes ePC functionally closer to backprop. A clearer explanation of how ePC is (can be) actually different from backprop would strengthen the paper’s overall contribution.

The authors test ePC on several neural network models and compare it with standard predictive coding methods and include training speed, number of inference steps, and model accuracy.

Most claims are supported by experiments on the evaluated benchmarks. However, the evaluation focuses mainly on standard image classification datasets, which limits broader conclusions. The experimental evaluation was done on standard image classification benchmarks such as MNIST, FashionMNIST, and CIFAR-10/100. While these datasets are appropriate for benchmarking against prior PC work and clearly demonstrate the scaling advantage of ePC over sPC, they do not fully explore domains for larger scale datasets.

(most critical weakness) The paper should also need to discuss the "digital hardware". Platform details, and more performance & hardware-related discussions/tests are missing.

The authors theoretically show that ePC produces the same learning updates as traditional predictive coding at equilibrium. They also show that ePC allows faster convergence because errors are propagated using backpropagation instead of iterative inference.

---

> ### Author Rebuttal · Authors · 2026-03-30
>
> We thank the reviewer for their very positive reception of the paper and their kind praise for the timeliness of our work and the manuscript's form and clarity.
>
> ‎
>
> ## About digital hardware
> The reviewer asks about the digital hardware implications of ePC's use cases.
>
> In this work, we focus exclusively on the digital *simulation* of PC. This comes without implications for the underlying hardware, which may be any general-purpose digital computer. The exact details (CPU, RAM, GPU) of the platforms we used in our experiments, can be found in Appendix E.
>
> The "digital hardware" mention comes from the central argument of the paper: PC suffers from a hardware-algorithm mismatch. sPC is meant for analog, not digital. We fill this algorithmic gap with ePC, a PC algorithm designed for digital (but not analog). Unlike sPC, ePC is intended for conventional computers and does not require custom digital hardware to function properly.
>
> To better reflect this generality in digital hardware, we suggest altering the paper's title, from "for Digital Hardware" to "in Digital Simulation", as well as adding a clarification to the introduction.
>
> ‎
>
> ## Differences between ePC and backprop
>
> The reviewer acknowledges that ePC produces exact-PC weight gradients, but still perceives a functional similarity with backprop. This reveals a tension: ePC *looks* like it's backprop, but it's clearly not. How can that be?
>
> To compare backprop and ePC, it's important to distinguish between the *optimization problem* and the *solver method*.
>
> Backpropagation is a general numerical solver method, that can be used to minimize any metric with respect to some target variables. In conventional Deep Learning ("backprop"), this metric is the loss, and the target variables are the model parameters. In ePC, the metric is the PC energy, and the target variables are the PC errors.
>
> So yes, the solver is the same, but the underlying optimization problem is very different. This is why backprop and ePC produce different weight gradients, as shown in Figure C.1.
>
> We will add this explanation to Appendix C.4, which is dedicated to the comparison between ePC and backprop.
>
> ‎
>
> ## Further benchmarking of ePC
>
> We agree that additional experiments could further strengthen the paper. Importantly, as the reviewer notes, our current set of experiments is appropriate and sufficient to support our main claims of ePC's superiority over sPC.
>
> As stated in Section 5, our experimental scope deliberately mirrors the setup of Pinchetti et al. (2025), an ICLR Spotlight benchmark specifically established to study the research question of why sPC doesn't scale. This allows for fair comparison with prior PC methods, which struggle not necessarily with large-scale datasets, but rather with deep architectures like ResNet-18.
>
> ‎
>
> ## Conclusion
>
> We appreciate the reviewer's support for our paper and will incorporate their suggestions to better delineate our digital hardware perspective and further differentiate ePC from backprop. With our rebuttal, we hope to have strengthened the reviewer's conviction of the merits of our work and its qualification for acceptance.

---

> > ### Author Rebuttal · Reviewer_Yknt · 2026-04-04
> >
> > Thanks for the explanations; things are getting clearer on the reviewer's end. I am supportive of the paper's acceptance and increasing my scores.

---

### Decision · Program_Chairs · 2026-04-30

**Decision:**

Accept (regular)

**Comment:**

The paper proposes error-based Predictive Coding (ePC), addressing a key limitation of standard predictive coding (sPC), namely exponential signal decay in digital simulations, and demonstrates strong empirical improvements in scalability and efficiency.

Reviewers agree that the problem is important and that the proposed method is technically sound, well presented, and effective in overcoming the core limitation of sPC.

The main concerns relate to the conceptual positioning of ePC, in particular its reliance on backpropagation-like mechanisms, the reduced biological plausibility, and the unclear advantage over standard backpropagation. Some reviewers also note limited experimental scope and questions regarding hardware implications.

However, the rebuttal clarified several of these points, and two reviewers indicated that their concerns were fully addressed.

Overall, I find the contribution meaningful, particularly in advancing predictive coding methods toward scalable settings, and the remaining concerns do not outweigh the strengths.